# Competition between hematopoietic stem and progenitor cells controls hematopoietic stem cell compartment size

Runfeng Miao[1], Harim Chun[2], Xing Feng[1], Ana Cordeiro Gomes[1,4], Jungmin Choi [2,3] ✉ & João P. Pereira [1] ✉

Cellular competition for limiting hematopoietic factors is a physiologically regulated but poorly understood process. Here, we studied this phenomenon by hampering hematopoietic progenitor access to Leptin receptor+ mesenchymal stem/progenitor cells (MSPCs) and endothelial cells (ECs). We show that HSC numbers increase by 2-fold when multipotent and lineage-restricted progenitors fail to respond to CXCL12 produced by MSPCs and ECs. HSCs are qualitatively normal, and HSC expansion only occurs when early hematopoietic progenitors but not differentiated hematopoietic cells lack CXCR4. Furthermore, the MSPC and EC transcriptomic heterogeneity is stable, suggesting that it is impervious to major changes in hematopoietic progenitor interactions. Instead, HSC expansion correlates with increased availability of membrane-bound stem cell factor (mSCF) on MSPCs and ECs presumably due to reduced consumption by cKit-expressing hematopoietic progenitors. These studies suggest that an intricate homeostatic balance between HSCs and proximal hematopoietic progenitors is regulated by cell competition for limited amounts of mSCF.

In adult mammals, hematopoietic stem cells (HSCs) rely on a combination of key paracrine signals provided by specialized microenvironments in the bone marrow and by the liver. While HSCs can access long-range acting signals, such as hepatocyte-produced Thrombopoietin[1], presumably anywhere, other signals such as Stem Cell Factor (SCF, or Kit ligand encoded by *Kitl*) and CXCL12 are preferentially accessed in specialized niches formed predominantly by perisinusoidal mesenchymal stem/progenitor cells (MSPCs) and endothelial cells (ECs) in the bone marrow[2–9]. Although historically defined as hematopoietic stem cell niches, more recent studies demonstrated that these niches are also critical for the development of B-lymphoid lineage cells due to the fact that MSPCs and ECs are also exclusive cellular sources of IL7, a key lymphopoietic cytokine[10–13]. Besides IL7, CXCL12, and SCF, MSPCs and ECs also express several key

hematopoietic cytokines with well-defined roles in myeloid and lymphoid-lineage cell differentiation, such as FLT3L, MCSF, IL34, IL15, GCSF among others, as revealed by single-cell RNA sequencing of non-hematopoietic bone marrow cell populations[14–16]. Importantly, common myeloid and lymphoid progenitors, and megakaryocyte and erythroid progenitors depend on SCF produced by Lepr+ MSPCs, whereas macrophage and dendritic cell precursors and monocytes depend on MCSF produced by ECs in bone marrow[17–19]. Combined, these studies led us to propose that MSPCs and at least some ECs are not only required for the long-term maintenance of HSCs but also form appropriate environments for the development of most, if not all, lymphoid and myeloid cell lineages[20]. Terminally differentiated hematopoietic cell subsets, namely macrophages, megakaryocytes, and regulatory T cells (Tregs), can in turn relay signals such as

[1]Department of Immunobiology and Yale Stem Cell Center, Yale University School of Medicine, 300 Cedar Street, New Haven, CT 06519, USA. [2]BK21 Graduate Program, Department of Biomedical Sciences, Korea University College of Medicine, Seoul 02841, Republic of Korea. [3]Department of Genetics, Yale University School of Medicine, 300 Cedar Street, New Haven, CT 06519, USA. [4]Present address: i3S – Instituto de Investigação e Inovação em Saúde, University of Porto, Porto, Portugal. ✉e-mail: jungmin.choi@yale.edu; joao.pereira@yale.edu

adenosine, TGFβ, and PF4 back to the HSC, or alter HSC niche activity through changes in CXCL12 production, and control the size of the HSC compartment under homeostatic conditions[21–24].

While some hematopoietic cytokines produced by MSPCs and ECs act in cell lineage-restricted manners (e.g., MCSF in monocyte/macrophage development; IL7 in lymphoid lineage development, etc.), other signals such as SCF are shared by multiple hematopoietic progenitor subsets. This type of cellular organization in which MSPCs and ECs harbor a constellation of hematopoietic cells and nurture distinct cell lineages raises the possibility that competition between HSCs and downstream progenitors for common and limiting resources could control HSCs and hematopoietic progenitors under homeostasis and during perturbations. However, arguments have been made against this possibility. Specifically, MSPCs and ECs outnumber HSCs by more than 10 fold, indicating that many putative HSC niches may remain "vacant"[25,26]. But, when taking into account not only the number of HSCs but also of downstream progenitors (e.g., CMPs, CLPs, MEPs, GMPs, etc.), the stoichiometry between the number of niche cells and of hematopoietic stem and progenitor cells is lower than one.

In this work, we tested whether cellular competition between HSCs and hematopoietic progenitors for HSC niche factors could control the HSC compartment size. HSCs and hematopoietic progenitor and differentiated cells utilize the CXCR4/CXCL12 pathway for bone marrow homing, and for access to the marrow parenchyma where they are retained via adhesive interactions with CXCL12-producing MSPCs and ECs. By examining the impact of CXCR4 conditional deletion at multiple stages of hematopoietic cell development, we find that when MPPs are deficient in CXCR4, the HSC compartment size increases by ~ 2-fold without any measurable loss of HSC fitness. This increase occurs without any major changes in the MSPC and EC transcriptome nor in transcriptional heterogeneity of the non-hematopoietic compartment. Surprisingly, HSC expansion is entirely controlled by excess membrane-bound SCF (mSCF) on MSPCs and ECs due to its reduced consumption by cKit-expressing hematopoietic progenitor cells. These studies provide insights into the fine-balance between HSCs and downstream progenitors regulated by a poorly understood phenomenon of cellular competition in the hematopoietic organ.

## Results

### HSC homeostasis requires CXCR4 in progenitor cells

In prior studies, we noted that HSCs and MPPs could be found in close proximity to each other and to the same bone marrow niche cell[10], suggesting that individual niche cells support a variety of hematopoietic stem and progenitor cells. Consistent with this possibility, recent studies demonstrated that besides HSCs, lymphoid, myeloid, and erythroid precursors also require SCF produced by Lepr+ MSPCs[17–19]. These studies led us to ask if competition between HSCs and downstream hematopoietic progenitors for factors locally produced by niche cells could control the HSC compartment size. Most signals produced by niche cells act in a short-range manner, and access to such signals is dependent on localization cues of which CXCL12 is the most abundantly produced. Thus, we analyzed mice in which HSCs express CXCR4 and respond to its ligand CXCL12 while MPPs and downstream hematopoietic progenitor and differentiated cells lack CXCR4 via conditional deletion using *Flk2*-driven Cre recombinase (Fig. 1a). *Flk2*-cre has been described to efficiently target MPPs and downstream hematopoietic cells leaving HSCs unchanged[27]. We found that phenotypic long-term HSC numbers increase by ~ 2-fold in the bone marrow, while short-term HSCs and MPP2 and MPP3 cell populations[28] remain unchanged (Fig. 1b, c, and Supplementary Fig. 1a, b). The number of lymphoid-primed MPP4 cells is dramatically reduced in bone marrow (Fig. 1c and Supplementary Fig. 1a, b) and increased in the spleen (Fig. 1d), consistent with a critical role for CXCR4 in HSC and hematopoietic progenitor cell retention in bone marrow[10,29]. We also observe increased numbers of phenotypic HSCs in the spleen (Fig. 1d), suggesting an overall increase in medullary and extra-medullary HSCs. The number of LSKs in bone marrow is unchanged (Fig. 1e). Other myeloid and lymphoid progenitors and differentiated immune cells are also significantly reduced in the bone marrow (Supplementary Fig. 1a, b), as expected[10]. To assess if the number of functional HSCs is increased, we analyzed their ability to

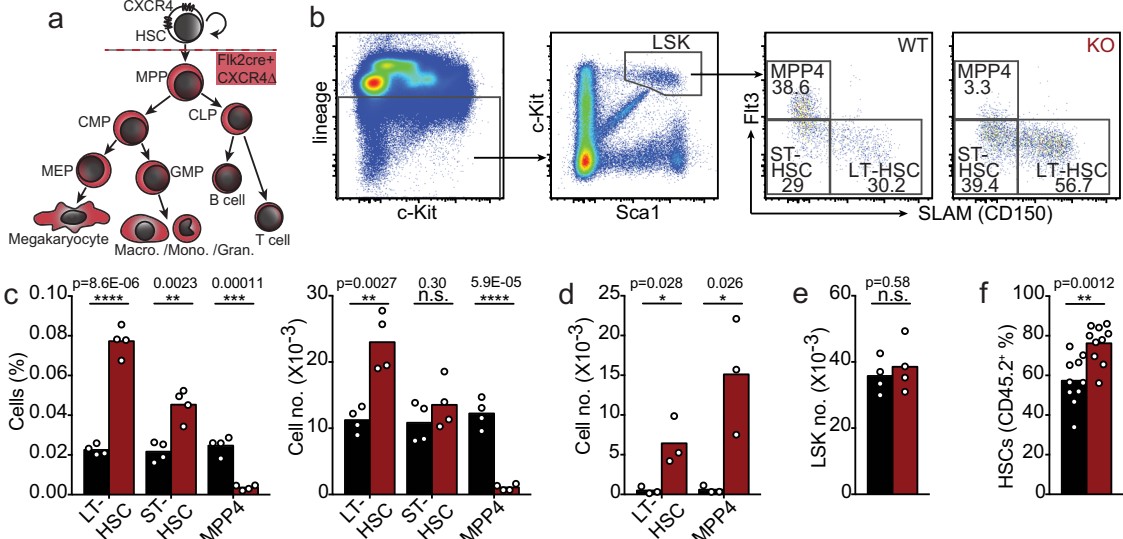

**Fig. 1 | Increased HSC numbers in mice with CXCR4-deficient MPPs. a** Schematic representation of CXCR4 deletion in hematopoietic multipotent progenitor cells with a *Flk2*-cre transgene. **b** Gating strategy for LSKs, LT-HSC, ST-HSC, and MPP4 cells. **c, d** LT-HSC, ST-HSC, and MPP4 cell numbers in bone marrow (**c**) and in spleen (**d**). **e** LSK cell number in bone marrow. In panels C-E, cells were collected from one femur and tibia, or from spleen, of *Flk2-cre.Cxcr4^{fl/+}* (CTR, black) and *Flk2-cre.Cxcr4^{fl/fl}* mice (cKO, red). **f** HSC chimerism in bone marrow of mice reconstituted with 50% CD45.2⁺ CTR or cKO bone marrow cells mixed with 50% CD45.1⁺ wild-type bone marrow cells. Data in all panels are representative of two or more experiments. Panels **c** and **e** are derived from n = 4 mice per group, panel **d** is from *n* = 3/group, and panel F is derived from 10 mice per group. Bars indicate average, circles depict individual mice. *P < 0.05; **P < 0.01; ***P < 0.001; and ****P < 0.0001 by unpaired two-sided Student's *t* test. Source data are provided as a Source Data file.

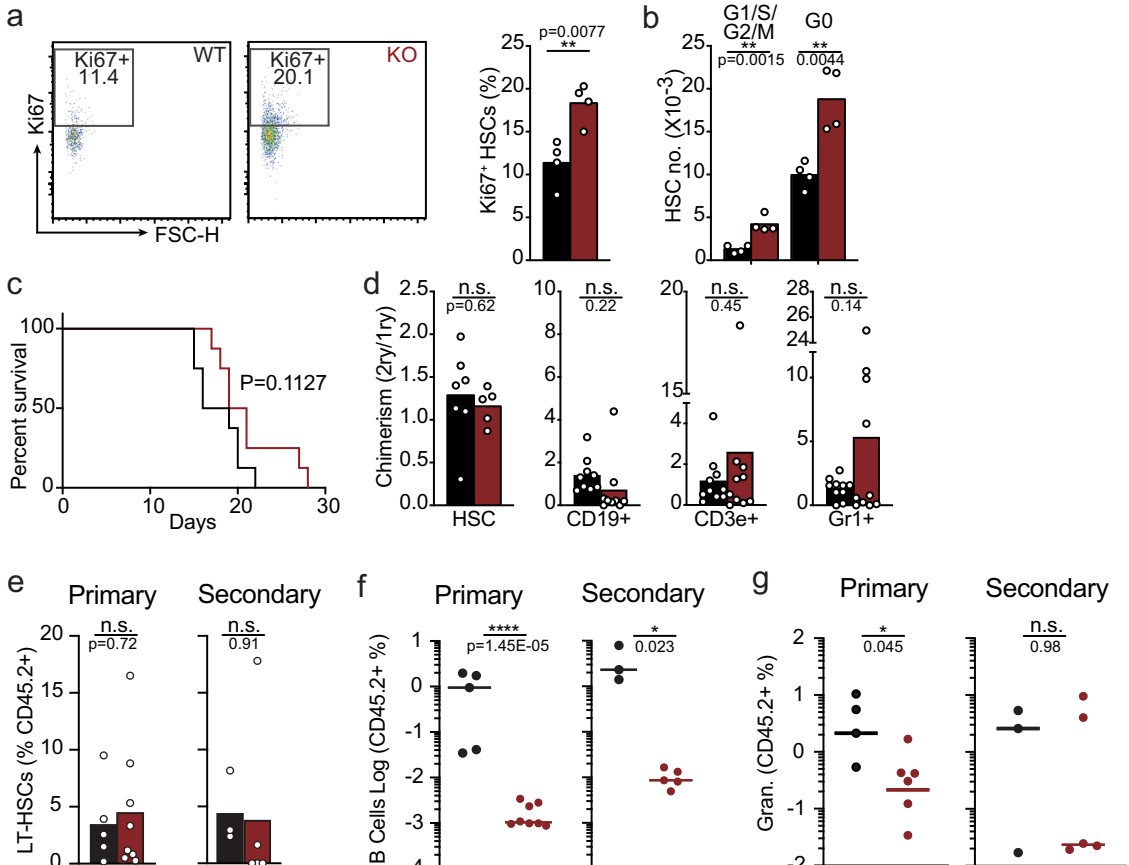

**Fig. 2 | Measurement of HSC quiescence and self-renewal in mice with CXCR4-deficient MPPs. a**, **b** HSC cell cycle status. **a** Gating strategy and cell frequency (*n* = 4 mice/group); (**b**) Enumeration of HSCs in G0 (Ki67⁻) and in G1/S/G2M (Ki67⁺) in bone marrow (*n* = 4 mice/group). **c** Kaplan–Meyer survival plot of mice following weekly 5-FU treatment (i.p., 150 mg/kg, *n* = 8 per group). **a**–**c** data obtained from *Flk2-cre.Cxcr4*ᶠˡ/⁺ (black) and *Flk2-cre.Cxcr4*ᶠˡ/ᶠˡ (red) mice (**d**) Serial transplantation: Ratio of bone marrow chimerism in HSCs displayed as chimerism in secondary transplantation divided by chimerism in primary transplantation. Mice were reconstituted with 50% CD45.2⁺ CTR or cKO bone marrow cells mixed with 50% CD45.1⁺ wild-type bone marrow cells (*n* = 10/group). **e**–**g** Long-term self-renewal and differentiation potential of 30 phenotypic HSCs analyzed during primary and secondary transplantation (*n* = 5 and *n* = 8). **e** HSC chimerism; (**f**) CD19+ B cell chimerism; (**g**) Gr1+ granulocyte chimerism. Data in all panels are representative of two or more experiments. Bars and lines indicate average, circles depict individual mice. n.s. not significant; *P* > 0.05 by unpaired two-sided Student's *t* test. Source data are provided as a Source Data file.

re-populate the HSC compartment of lethally irradiated recipient mice. In a setting of mixed bone marrow transplantation with 50% bone marrow cells from CXCR4 conditionally deficient mice (*Flk2*-cre⁺ *Cxcr4*ᶠˡ/ᶠˡ, from here on referred to as cKO) and 50% bone marrow cells from wild-type (WT) C57BL6/NCI mice distinguished by CD45 isoforms (CD45.1 and CD45.2), the mixed chimerism in the HSC compartment of cKO:WT chimeric mice is significantly higher than that in control WT:WT mixed chimeras 16 weeks after hematopoietic reconstitution (Fig. 1f). In sharp contrast, the mixed chimerism in MPP4s and in differentiated hematopoietic cells is dramatically reduced in cKO:WT chimeric mice (Supplementary Fig. 1c), as expected[10]. Collectively, these data reveal that phenotypic HSCs become numerically increased when MPP4 and downstream hematopoietic progenitors lack CXCR4.

## Effects on HSC quiescence and self-renewal

Increased numbers of phenotypic HSCs in cKO mice suggests increased entry into cell cycle, and possibly reduced quiescence. To test this, we analyzed intracellular levels of the nuclear protein associated with cell proliferation Ki67 in phenotypic HSCs of both mice by flow cytometry. Although we find an increased frequency of Ki67⁺ HSCs in cKO mice (Fig. 2a), this did not result in reduced numbers of Ki67⁻ quiescent HSCs (Fig. 2b). Furthermore, when challenging mice with the myelosuppressive agent 5-Fluorouracil (5-FU), cKO mice are equally resistant to treatment as WT littermate controls (Fig. 2c), which

contrasts sharply with increased susceptibility to 5-FU when HSCs and downstream hematopoietic cells lack CXCR4[30]. To determine if phenotypic HSCs are functionally normal, we examined their capacity for long-term self-renewal and multilineage differentiation in vivo. In transplantation experiments of 50% mixed bone marrow chimeras (50% cKO or littermate and 50% CD45.1+ C57BL6), the mixed chimerism of HSCs, B lymphocytes and granulocytes from cKO bone marrow remains stable after 16 weeks of primary bone marrow transplantation followed by another 16 weeks of secondary transplantation (Fig. 2d). These results show that HSCs in cKO mice are functionally equivalent to HSCs from control littermate mice. To specifically determine if phenotypic HSCs are functionally normal, we sorted HSCs from cKO mice or control littermate (30 phenotypic HSCs) and transplanted into lethally irradiated C57BL6 (CD45.1+) recipient mice, followed by secondary transplantation into new cohorts of C57BL6 (CD45.1+) recipient mice. The mixed chimerism in the phenotypic HSC, CD19+ B cell, and Gr1+ granulocyte compartments were determined 16 weeks after the primary and secondary transplant. HSCs from WT and cKO mice are equivalent in their ability to self-renew and to differentiate into lymphoid and myeloid-lineage cells upon primary and secondary transplantation (Fig. 2e–g). Consistent with these observations, HSCs isolated from cKO and WT mice display similar expression of CD150 or CD41 on the cell surface (data not shown), markers whose increased expression is associated with myeloid differentiation bias[31,32].

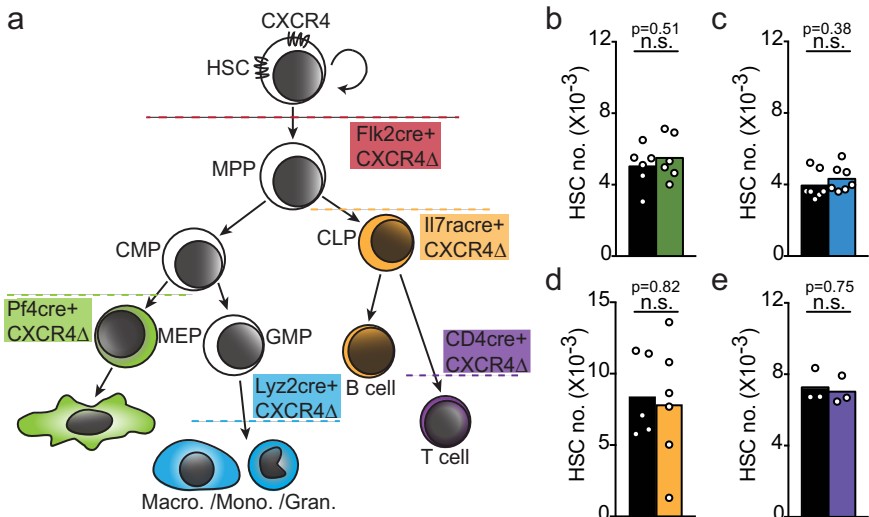

**Fig. 3 | Normal HSC numbers in mice carrying *Cxcr4* deletion in megakaryocytes, myeloid, and lymphoid cells. a** Schematic representation of CXCR4 deletion in hematopoietic cells by cell-lineage specific cre-recombinase transgenic approaches. **b**–**e** LT-HSC numbers in one femur and tibia of control littermate mice (black); (**b**) *Pf4-cre.Cxcr4^{fl/fl}* (green, *n* = 6/group); (**c**) *Lyz2-cre.Cxcr4^{fl/fl}* (blue, *n* = 7/group); (**d**) *Il7ra-cre.Cxcr4^{fl/fl}* (yellow, CTR *n* = 5, KO *n* = 6); (**e**) *Cd4-cre. Cxcr4^{fl/fl}* (purple, *n* = 3/group). Data are representative of two or more experiments. Bars indicate average, circles depict individual mice. n.s. not significant, *P* > 0.05 by unpaired two-sided Student's *t* test. Source data are provided as a Source Data file.

Combined, these studies show a numerical increase in phenotypic and functional HSCs in cKO mice without loss of HSC quiescence, self-renewal, or differentiation bias.

## CXCR4 in immune cells and effects on HSCs

The HSC compartment size is controlled by approximately 2-fold through feedback signals such as TGFβ, PF4, and adenosine, provided by differentiated hematopoietic cells[21–24]. *Cxcr4* deletion at the MPP stage ablates CXCR4 function in differentiated hematopoietic cells that control HSC numbers, namely megakaryocytes, macrophages, and Tregs[21–24]. To test if differentiated hematopoietic cells require CXCR4 for controlling the size of the HSC compartment, we conditionally deleted *Cxcr4* in differentiated immune cells downstream of the MPP stage (Fig. 3a) using multiple Cre-recombinase transgenic approaches (Abram et al., 2014; Schlenner et al., 2010; Tiedt et al., 2007). We find no measurable differences in HSC numbers, frequency, and cell cycle status in the bone marrow of mice in which *Cxcr4* is deleted in megakaryocytes (*Pf4*-cre; Fig. 3b and Supplementary Fig. 2a), macrophages (Lyz2-cre; Fig. 3c and Supplementary Fig. 2b), lymphoid cells (*Il7ra^{Cre/+}*; Fig. 3d and Supplementary Fig. 2c), and T cells (*Cd4*-cre; Fig. 3e and Supplementary Fig. 2d). In mice conditionally deficient in CXCR4 in T cells, the number of bone marrow Tregs is significantly reduced and to a greater extent than in MPP CXCR4 cKO mice (Supplementary Fig. 2e). In mice conditionally deficient in CXCR4 in megakaryocytes or in myeloid cells, the number of bone marrow megakaryocytes, neutrophils, monocytes, and CD169+ macrophages is equivalent to that in control littermate mice (Supplementary Fig. 2f, g). The total bone marrow and spleen cell number of the multiple CXCR4 cKO mice is equivalent to that of control littermates, except for MPP CXCR4 cKO mice that show reduced BM cell numbers (Supplementary Fig. 2H–l), as expected[10]. Thus, the increased numbers of HSCs seen in mice in which MPPs lack *Cxcr4* is not due to the lack of CXCR4 expression in differentiated hematopoietic cells, including in Tregs[24].

## Stability of HSC niche cell transcriptional heterogeneity

The ability of MSPCs and ECs to produce hematopoietic cytokines and chemokines can be altered under pre-leukemic and leukemic states[12–14,20]. To determine if the HSC niche is altered when MPPs lack *Cxcr4*, we analyzed the transcriptional heterogeneity of non-hematopoietic bone marrow cells (Supplementary Fig. 3a) by droplet-based single-cell RNA sequencing (scRNA-seq). A total of 14,027 cells, of which 6413 were from control and 7614 from *Cxcr4* cKO mice, were profiled at a mean depth of 39,451 and 51,660 reads/cell, respectively. After quality control utilizing Seurat (Stuart and Butler et al., 2019) (see Methods) and removal of contaminating hematopoietic cells, we analyzed 1380 control and 2438 cKO cells for a total of 3818 non-hematopoietic cells. We used the dimensional reduction technique Uniform Manifold Approximation and Projection (UMAP) to visualize non-hematopoietic cell clusters and compare differences in cell cluster heterogeneity between control and cKO bone marrow samples[33]. Unsupervised clustering identified two mesenchymal lineage cell clusters marked by *Lepr* expression, three osteolineage clusters, four endothelial cell clusters, one fibroblast cell cluster, and one chondrocytic cell cluster in both datasets (Fig. 4a, b, and Supplementary Fig. 3b, c). The total number of stromal and endothelial cell clusters resolved is reduced when compared with those described in prior studies[14–16], possibly because of a reduced number of cells sequenced. Nevertheless, the overall structure of the data is comparable: one cycling EC cluster (cluster 11), two major cell clusters representing arterial and arteriolar ECs (clusters 10 and 2, respectively), and one sinusoidal EC subset (cluster 1); a large population of Lepr+ mesenchymal lineage cells (cluster 0); and a small cluster of Lepr+ mesenchymal lineage cells with increased expression of immediate early genes (e.g., *Fos*, *Fosb*, *Jun*, *Nr4a1*, *Mcl1*, cluster 7)[34]. Although cluster 9 is transcriptionally similar to cluster 1, cluster 9 cells express very low amounts of EC-specific genes (Fig. 4b shows *Cdh5*, but similar results could be seen for *Kdr*, *Flt4*, and *Flt1* expression). Comparison of cell clusters identified in control and cKO bone marrow samples reveal considerable overlap between samples (Fig. 4c), and a comparable proportion of individual clusters (Fig. 4d). Differential abundance test between control and cKO datasets using Milo[35] confirm that the abundance of cellular states is not substantially different in all clusters (Fig. 4e; spatial False Discovery Rate (FDR) < 0.05). Furthermore, we find a very small number of differentially expressed genes (DEGs) between control and cKO mesenchymal and endothelial cell clusters. The major Lepr+ cell population (cluster 0) with an adipocytic gene expression program previously identified[16] showed 11 DEGs with LogFC <1 (Supplementary Data 1). Likewise, the major sinusoidal and

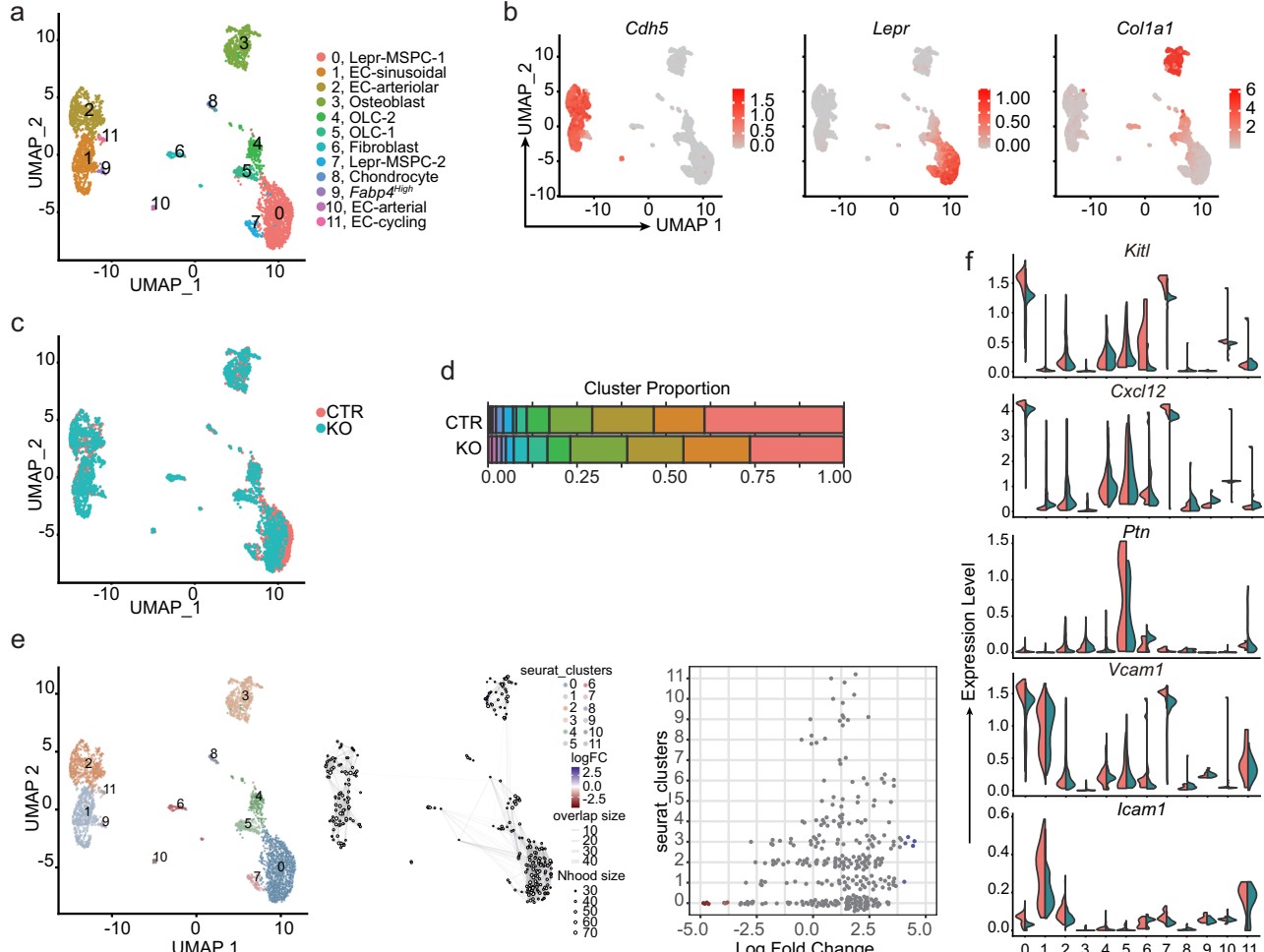

**Fig. 4 | Niche cell transcriptional heterogeneity in mice with CXCR4-deficient MPPs. a** UMAP visualization of bone marrow non-hematopoietic cells. **b** Expression levels of genes associated with endothelial cells (*Cdh5*), MSPCs (*Lepr*), and osteo-lineage cells (*Col1a1*) overlaid on UMAP. **c** Overlay of UMAP visualization of bone marrow non-hematopoietic cells from *Flk2-cre.Cxcr4*[fl/+] (CTR, red) and *Flk2-cre.Cxcr4*[fl/fl] (KO, blue) mice. **d** Cluster proportion. **e** Differential abundance test with Milo. UMAP cluster representation (left); graph representation of Milo differential abundance testing (middle); Beeswarm plot showing the distribution of logFC in neighborhoods containing cells from different Seurat clusters (right). Nodes are neighborhoods colored by logFC between CTR and KO. Non-differential abundance neighborhoods (FDR 5%) are colored white; neighborhood sizes correspond to number of cells in each neighborhood. Graph edges represent the number of cells shared between adjacent neighborhoods. Cell cluster frequencies in each sample (horizontal-colored bars). **f** Violin plots representing expression levels of essential HSC regulators (*Kitl, Cxcl12, Ptn, Vcam1, Icam1*) in CTR (red) and KO (blue) cell clusters.

arteriolar EC clusters reveal 23 and 11 DEGs with LogFC <1, respectively (Supplementary Data 1). Mesenchymal and osteolineage cells (clusters 3-7, Supplementary Fig. 3d) also reveal <15 DEGs in which differences between samples did not exceed two-fold (Supplementary Data 1). Importantly, none of the DEGs are implicated in HSC regulation nor in downstream hematopoietic progenitor differentiation, and no major biological pathway can be associated with DEGs revealed in these analyses. Major regulators of HSC homeostasis such as *Cxcl12, Kitl, Ptn*, and adhesion ligands *Vcam1* and *Icam1*[4–6,30,36–40] are transcriptionally abundant and equivalent in mesenchymal and endothelial cell clusters of both samples (Fig. 4f). Furthermore, no differences are detected in hematopoietic growth factors expressed in both samples (Supplementary Fig. 3f). Finally, to test if the bone marrow environment of cKO mice can support HSCs we transplanted WT BM into lethally irradiated WT or cKO recipient mice. We found similar numbers of phenotypic HSCs in the bone marrow of WT and cKO recipient mice 6 weeks after transplantation (Supplementary Fig. 3g). Taken together, these studies suggest that major transcriptional changes in HSC niches could not explain the HSC expansion seen in cKO mice, and that the cKO bone marrow environment per se is insufficient for causing HSC expansion.

To increase the sensitivity of detecting DEGs, we performed RNA sequencing of bulk sorted MSPCs. This analysis revealed 53 DEGs (normalized RNA counts >0 in all samples, *q* value < 0.05) (Fig. 5a–c), of which a few are hematopoietic cytokines and chemokines (Fig. 5d). Of note, *Igfbp2* and *Igf2bp2* have been shown to play roles in HSC homeostasis[41,42], and their increased expression in MSPCs may have contributed to increased HSC numbers seen in cKO mice. Thus, bulk and scRNAseq studies revealed a remarkable stability of niche cell transcriptome and niche cell heterogeneity under conditions in which hematopoietic progenitors and differentiated cells interact poorly with CXCL12-producing niche cells.

## HSCs and hematopoietic progenitors compete for SCF

We considered the possibility that the abundance of hematopoietic growth factors displayed on WT and cKO niche cells may differ[8,43]. Of note, SCF is particularly attractive given its critical role in HSC homeostasis, survival, and proliferation, and the fact that mSCF is particularly important for hematopoiesis[44–46]. Although transcript levels for SCF (*Kitl*) did not reveal differences between cKO and control Lepr+ MSPCs (Fig. 6a), we detected a significant increase in mSCF on bone marrow Lepr+ MSPCs and ECs (Fig. 6b–d). We performed a series of

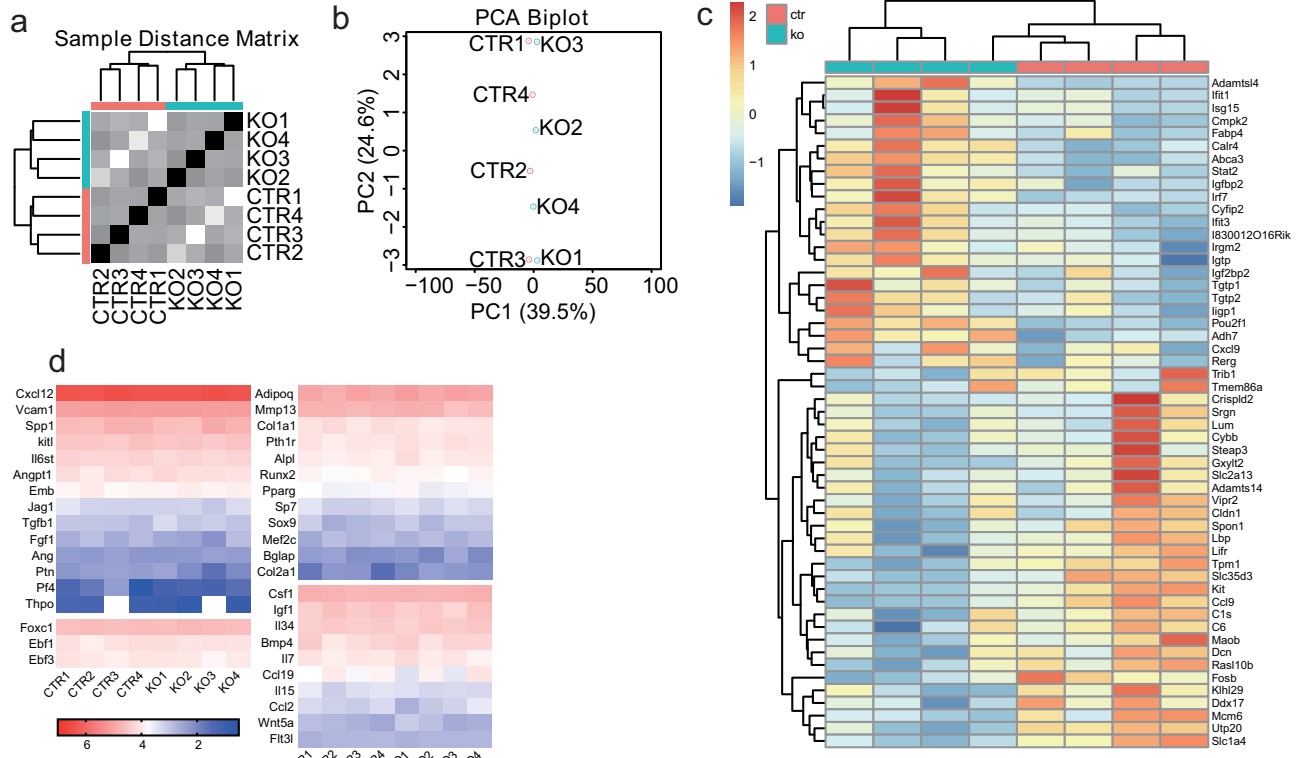

**Fig. 5 | Comparison of MSPC global transcriptome between control and mice with CXCR4-deficient MPPs. a** Unsupervised hierarchical sample clustering represented as Sample Distance Matrix. CTR (*Flk2-cre.Cxcr4$^{fl/+}$*) and KO (*Flk2-cre.Cxcr4$^{fl/fl}$*). **b** Principal-component analysis (PCA) representation of *Flk2-cre.Cxcr4$^{fl/+}$* (CTR) and *Flk2-cre.Cxcr4$^{fl/fl}$* (KO) samples obtained from individual mice.

**c** Differentially expressed genes in MSPCs isolated from *Flk2-cre.Cxcr4$^{fl/+}$* (red) and *Flk2-cre.Cxcr4$^{fl/fl}$* (blue) mice. Plotted genes showed normalized RNA counts >0 in all samples, FDR adjust $p < 0.05$. **d** Heatmap representation of log10 normalized RNA counts of genes important for HSC maintenance and multilineage differentiation. All panels are derived from $n = 4$ mice per group.

in vitro and in vivo experiments to validate the specificity of mSCF staining. *Kitl* overexpression in OP-9 stromal cells showed robust mSCF expression relative to control transduced OP9 cells (Supplementary Fig. 4a). Pre-incubation of the anti-SCF antibody with varying amounts of soluble SCF completely prevents mSCF detection on cKO MSPCs and ECs in a SCF dose-dependent manner (Supplementary Fig. 4b). However, we could not reliably detect mSCF on WT MSPCs when comparing to background staining obtained in *Kitl*-deficient MSPCs (Supplementary Fig. 4c). Importantly, reducing mSCF protein abundance by half through genetic means (i.e., crossing cKO mice with *Kitl$^{GFP/+}$* mice), reduces mSCF protein levels in niche cells of cKO mice to the level detected in niche cells isolated from WT mice (Fig. 6e), and also reduces the number of phenotypic HSCs to physiological levels (Fig. 6f). In contrast, *Kitl* haploinsufficiency has no impact on HSC numbers of *Flk2*-cre *Cxcr4$^{fl/+}$* mice (Fig. 6f). Conditional *Kitl* deletion from Lepr $^+$ MSPCs of *Cxcr4* cKO mice also reduces HSC numbers significantly even though we could not detect changes in mSCF levels (Supplementary Fig. 4d, 4e). The levels of mSCF on MSPCs and ECs of mice conditionally deficient in *Cxcr4* in T cells (*Cd4*-cre), myeloid cells (*Lyz2*-cre), or in megakaryocyte (*Pf4*-cre) cells are normal (Supplementary Fig. 4f, g). The total numbers of Lepr $^+$ MSPCs and ECs are also equivalent between cell-lineage specific *Cxcr4* cKO mice and littermate controls (Supplementary Fig. 4h, i). Besides SCF, HSCs also require hepatocyte-produced Thrombopoietin[1]. Interestingly, we measured a small but significant reduction in Thrombopoietin concentration in the serum of cKO mice (Supplementary Fig. 5a). However, heterozygous mutations in *Thpo* do not change the total number of HSCs in vivo, suggesting that long-range acting Thrombopoietin is not limiting the size of the HSC compartment[1].

cKit signaling activates the JAK/STAT pathway and induces STAT3 and STAT5 phosphorylation[47,48]. HSCs stimulated with soluble SCF

in vitro for 30 minutes reveals a low but significant increase in pSTAT3 and pSTAT5 measured by intracellular phospho-flow cytometry (Supplementary Fig. 5c). Importantly, pSTAT3 but not pSTAT5 is significantly increased in HSCs isolated from the bone marrow of *Cxcr4* cKO mice relative to that detected in control littermate (Fig. 6g). Combined, these data lend support to the possibility that the increased HSC number seen in cKOs is dependent on increased mSCF availability on Lepr $^+$ MSPCs and ECs in the bone marrow.

Next, we aimed at understanding the mechanism underlying increased mSCF levels in HSC niche cells of cKO mice. Measurements of soluble SCF shows that cKO and WT mice have similar SCF concentrations in the bone marrow interstitial fluid (Supplementary Fig. 5b), suggesting that the increased mSCF level detected is not due to reduced proteolytic cleavage from niche cells. Therefore, we considered the possibility that competition between HSCs and cKit$^+$ hematopoietic progenitors might limit mSCF abundance in niche cells such that it influences the HSC compartment size. To test this possibility, we depleted cKit$^+$ cells in vivo by administering saturating amounts of an anti-cKit antibody (clone ACK2, Supplementary Fig. 5d) that blocks cKit signaling in vivo and elicits antibody-dependent cellular phagocytosis[49], and measured changes in mSCF displayed on MSPCs and ECs in the bone marrow. Treated animals show a ~6–8 fold reduction in cKit+ cells 3 days after the treatment (Fig. 6h) whereas neutrophil numbers reduce by only 20% (Supplementary Fig. 5e), as expected[50]. Remarkably, mSCF displayed on the surface of bone marrow MSPCs increases promptly in ACK2-treated animals relative to that detected in mice treated with isotype control antibody (Fig. 6i). To further assess if cKit+ progenitors actively consume SCF we blocked cKit signaling in vivo for 24 h by administering ACK2 in combination with BrdU i.v. and measured the rate of cKit+ progenitor cell generation by BrdU incorporation. The number of BrdU+ LSKs,

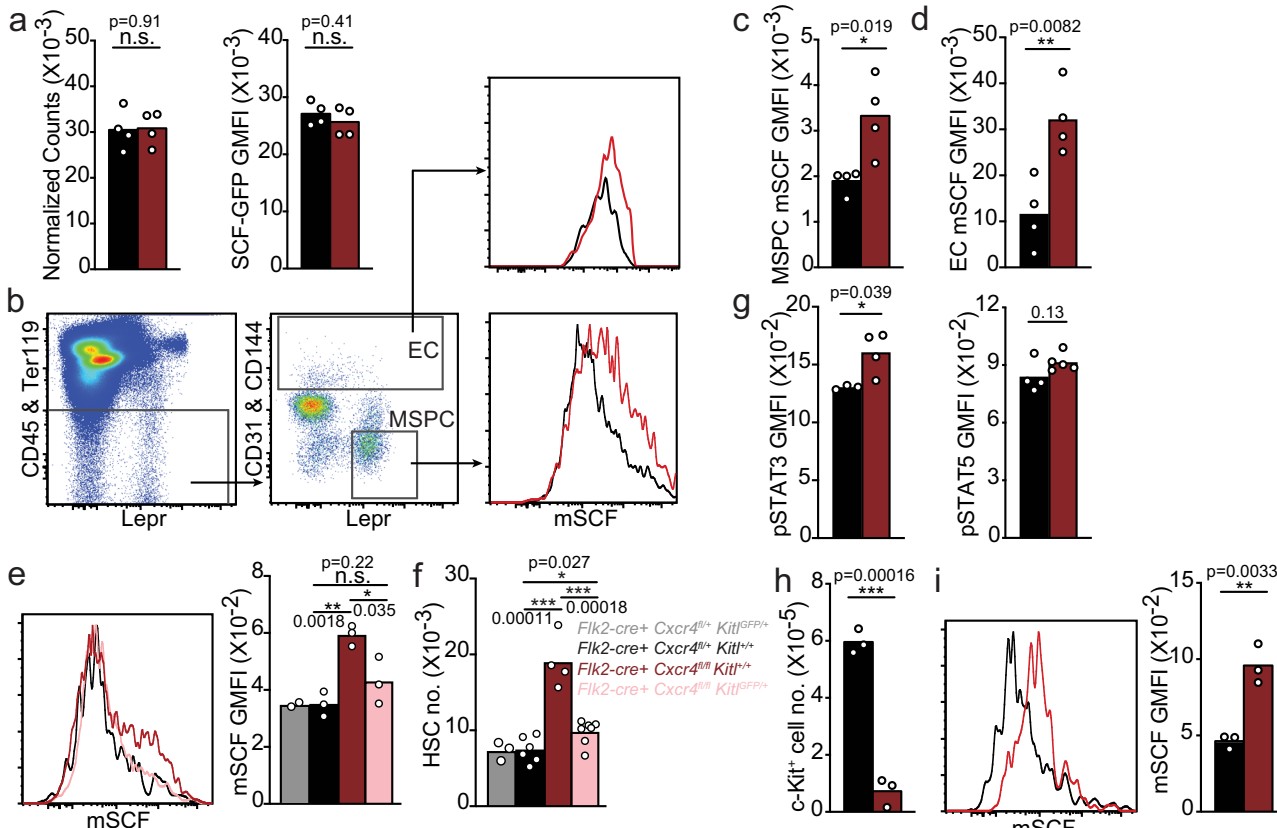

**Fig. 6 | Altered membrane-bound SCF on MSPCs and ECs, and altered cKit signaling in HSCs of mice with CXCR4-deficient MPPs. a** *Kitl* mRNA expression. Normalized *Kitl* RNA counts in MSPCs (left); and *Kitl*-GFP expression in gated LEPR⁺ MSPCs (right) of *Flk2-cre.Cxcr4^fl/+ Kitl^GFP/+* (black), and *Flk2-cre.Cxcr4^fl/fl Kitl^GFP/+* (red) mice (*n* = 4 mice/group). **b** Gating strategy of LEPR⁺ MSPCs and ECs, and mSCF measurements by flow cytometry. **c, d** mSCF geometric mean fluorescent intensity (GMFI) on LEPR⁺ MSPCs (**c**) and ECs (**d**) from *Flk2-cre.Cxcr4^fl/+* (black) and *Flk2-cre.Cxcr4^fl/fl* (red) mice (*n* = 4 mice/group). **e** Overlay of mSCF on gated LEPR⁺ MSPCs isolated from indicated mice (left), mSCF quantification (right, *n* = 3 mice/group). **f** HSC numbers in bone marrow. Gray, *Flk2-cre.Cxcr4^fl/+Kitl^GFP/+* (*n* = 3) Black, *Flk2-*

*cre.Cxcr4^fl/+Kitl^+/+* (*n* = 6); Red, *Flk2-cre.Cxcr4^fl/fl Kitl^+/+* (*n* = 4); Pink, *Flk2-cre.Cxcr4^fl/fl Kitl^GFP/+* (*n* = 7). **g** pSTAT3 (left) and pSTAT5 (right) levels in HSCs isolated from *Flk2-cre.Cxcr4^fl/+* (black, pSTAT3 *n* = 3, pSTAT5 *n* = 4) and *Flk2-cre.Cxcr4^fl/fl* (red, pSTAT3 *n* = 4, pSTAT5 *n* = 5) mice. **h** cKit⁺ cell number in bone marrow (**i**) mSCF levels on LEPR⁺ MSPCs (*n* = 3/group). (H and I) Mice were treated with 200 μg of isotype (black) or ACK2 antibody (red) for 3 days (*n* = 3/group). Data in panel (**a**) are from one bulk RNA sequencing experiment with *n* = 4 samples/group. Data in panels (**b**–**i**) are representative of two or more experiments. Bars indicate average, circles depict individual mice. *$P < 0.05$; **$P < 0.01$ and ***$P < 0.001$ by unpaired two-sided Student's *t* test. Source data are provided as a Source Data file.

CMPs, and MEPs is significantly reduced whereas BrdU+ CLPs are increased (Supplementary Fig. 5f–j). BrdU+ GMP numbers trend towards being reduced but the difference was not statistically significant (Supplementary Fig. 5h). These data demonstrate that cKit+ progenitors actively consume SCF and reveal an inverse relationship between mSCF abundance on niche cells and the number of cKit⁺ hematopoietic stem and progenitor cells in the bone marrow. Combined, these data suggest that mSCF acts as a carrying capacity factor that limits the size of the HSC compartment.

Finally, to gain insight into how cKit+ progenitor interactions with SCF-producing niche cells may result in mSCF consumption, we attempted to measure the extent to which hematopoietic cells physically remove membrane fragments and associated proteins from MSPCs, a process known as trogocytosis[51,52]. We generated mice that conditionally express membrane-bound GFP (mGFP) in mesenchymal-lineage cells (*Lepr^Cre/+ Rosa26* mT/mG[53]). These mice were lethally irradiated and reconstituted with non-fluorescent C57BL6/J BM cells in order to allow quantification of mesenchymal cell-derived membrane fragments (measured by mGFP fluorescence) latched on hematopoietic cell subsets. Interestingly, mGFP fluorescence is detected at highest levels on HSCs, with other cKit+ progenitors also revealing significant amounts of mGFP fluorescence (Supplementary Fig. 6a–c). In contrast, no significant mGFP fluorescence is measured on

inflammatory monocytes, whereas GMPs, recirculating mature B cells and neutrophils display minimal mGFP fluorescence (Supplementary Fig. 6b, c). To exclude the possibility that hematopoietic progenitors and differentiated immune cells acquire mGFP ex-vivo (i.e. during tissue processing for flow cytometry analyses) we performed parallel experiments in which we mixed non-fluorescent (CD45.2+) BM cells with BM cells isolated from *Lepr^Cre/+ Rosa26* mT/mG BM chimeras (CD45.1+). This experiment shows minimal acquisition of mGFP fluorescence in hematopoietic progenitors ex-vivo (Supplementary Fig. 6d). Combined, these findings lend support to the conclusion that active mSCF consumption by HSCs and hematopoietic progenitors limits mSCF availability which in turn stabilizes the HSC compartment size.

## Discussion

Guided by CXCR4, HSCs, hematopoietic progenitors, and differentiated immune cells localize in the bone marrow and physically interact with a heterogeneous population of cytokine and growth factor-producing mesenchymal and endothelial cells. While the lymphoid, myeloid, erythroid, and megakaryocyte lineages diverge in cytokine requirements at late stages of differentiation, they converge in their requirement for SCF during early developmental stages[20]. Studies over the last 10 years identified LEPR⁺ mesenchymal lineage

cells as relevant cellular sources of SCF for HSCs and lineage-restricted hematopoietic progenitors, including lymphoid, erythroid, and myeloid progenitors[4,17]. The LEPR+ MSPC population is transcriptionally and functionally heterogeneous and includes a major cell population characterized by an adipogenic gene expression program and smaller subsets of osteolineage-primed mesenchymal progenitors. Importantly, SCF produced by osteolineage-primed mesenchymal progenitors (marked by Osteolectin expression) contributes to CLP and early B and T cell progenitor development or maintenance, but it does not contribute to HSC homeostasis and early myeloid, erythroid, and megakaryocyte progenitor development in vivo[18]. Therefore, HSCs and multiple hematopoietic progenitors rely on SCF produced presumably by the same fraction of MSPC that is marked by high *Lepr*, *Kitl*, and *Cxcl12* expression, a model that is in agreement with earlier findings that HSCs and MPPs are occasionally found in close proximity to the same niche cell[10,11].

In this study, we uncovered an unexpected layer of regulation of HSC homeostasis. When CXCR4 is deleted in MPPs using a *Flk2*-cre allele, it renders essentially all downstream hematopoietic progenitors and differentiated hematopoietic cells CXCR4 deficient and unable to respond to CXCL12 produced by MSPCs and ECs. Prior studies have shown that hematopoietic stem and progenitor cells rely on CXCR4-mediated migration for cell-cell interactions with CXCL12-expressing niche cells[29,54]. Because CXCR4 deficiency results in reduced hematopoietic cell interactions with CXCL12 and SCF-producing MSPCs and ECs, we argue that it also results in reduced SCF consumption and thus increased mSCF availability. However, these studies do not exclude the possibility that additional factors not expressed by bone marrow niche cells may contribute to HSC expansion. The fact that the defects seen in HSC and in hematopoiesis in mice conditionally deficient in *Cxcl12* or *Kitl* in bone marrow niche cells is remarkably similar[4–6,55] further supports the possibility that CXCL12 acts primarily as a guidance cue for hematopoietic cells to localize in bone marrow and interact with MSPCs.

By displacing MPPs and downstream hematopoietic progenitors from CXCL12-producing niches in the bone marrow through conditional *Cxcr4* deletion, we made two unexpected observations. First, the HSC niche is transcriptionally stable and impervious to non-malignant cell-cell interactions. This finding contrasts with profound transcriptional changes that occur when niche cells interact with preB cell acute lymphoblastic leukemia or with acute myeloid leukemias, which impact hematopoiesis and MSPC differentiation in vivo[12–14,56]. These observations suggest that signals provided or triggered by leukemic cells are sensed by HSC niche cells and that such signals control hematopoietic cytokine and chemokine production. Second, HSCs and SCF-dependent cKit-expressing hematopoietic progenitors compete for limited amounts of membrane-tethered SCF in vivo. This model of limited SCF availability is supported by the fact that *Kitl* haploinsufficiency reduces the number of HSCs and hematopoietic progenitors by approximately 2-fold under homeostatic conditions[4,17].

Even though *Kitl* mRNA is abundantly expressed in MSPCs, we were unable to reliably detect mSCF protein on LEPR+ MSPCs of wild type mice when compared to the background staining detected in LEPR+ MSPCs of *Kitl* conditionally deficient mice (*Lepr-cre + Kitl*fl/fl; Supplementary Fig. 4c). However, mSCF protein was readily detected on MSPCs of wild type mice treated with a cKit blocking antibody (Fig. 6i). Furthermore, mSCF was also readily detected in MSPCs and ECs of mice in which hematopoietic progenitors interact poorly with niche cells (*Cxcr4* cKO mice; Fig. 6e and Supplementary Fig. 4d). Combined, these studies provide evidence supporting a model in which competition for limited amounts of mSCF establishes a fine balance between the HSC and cKit+ hematopoietic progenitor compartment size. This model is in agreement with earlier studies by Nagasawa and colleagues showing that when large numbers of

transplanted HSCs engraft into non-irradiated recipients, the number of host and donor-derived hematopoietic progenitors, such as GMPs, is reduced[25]. It is also consistent with recent findings showing discrepancies between LEPR+ MSPC *Kitl* transcript and SCF protein abundance[43], while providing an alternative explanation: cytokine consumption by hematopoietic progenitors may account for such discrepancies.

It is presently unclear why HSCs do not expand by more than 2-fold when mSCF levels are still elevated in *Cxcr4* cKO mice. It is interesting to note that prior studies examining the HSC niche and mechanisms controlling the HSC compartment size also reported ~2-fold changes. For example, Angiogenin and Embigin have been shown to limit HSCs numbers by ~2-fold[57]. Likewise, CXCL4 and TGFβ1 produced by megakaryocytes also limit HSC numbers by a similar magnitude. It is possible that the capacity for mSCF availability to modulate the HSC compartment size is limited by cues promoting HSC quiescence such as Angiogenin, Embigin, CXCL4 or TGFβ1. Future studies will determine if (and which) quiescence-promoting cues limit HSC growth under conditions of elevated mSCF availability.

The exact mechanism by which HSCs and hematopoietic progenitors consume mSCF remains unclear. Recent studies using intravital 2-photon microscopy revealed that HSCs are moderately dynamic under homeostasis[54,58]. Like developing B cells, HSC motility is dependent on active CXCR4 signaling, and HSC retention within bone marrow requires integrin-mediated adhesion[29,54]. Likewise, MPPs are also motile within bone marrow under transplantation, and also require CXCR4 for bone marrow retention[10,59]. It is possible that interactions between HSCs, hematopoietic progenitors, and SCF-producing niche cells result in the physical removal of mSCF through a process resembling trogocytosis[51,52]. Consistent with this model, we were able to measure, albeit indirectly, MSPC-derived membrane processes latched on HSCs and several hematopoietic progenitors, consistent with the idea that these cells actively interact and consume mSCF displayed on MSPCs. Future studies that allow visualization of mSCF on bone marrow niche cells may provide insights into the mechanism(s) that limit mSCF availability in vivo.

Cell competition for limiting resources is common and physiologically important. In the immune system, examples include B cell clonal competition during the germinal center reaction, competition for access to B lymphocyte survival cytokines such as BAFF, among several others[60,61]. In these cases, competition ensures the survival of the fittest cell or clone with direct benefit for the host (e.g., selection of high-affinity clones, elimination of autoreactive lymphocytes). Likewise, competition for limited amounts of mSCF may ensure that the fittest HSCs are maintained and contribute to blood cell production over time. Once HSCs divide and progressively differentiate into MPP and lineage-restricted progenitors, their shared dependency on mSCF may ensure homeostatic control of HSC and progenitor compartment sizes. These findings and model are more easily compatible with studies showing that HSCs actively contribute to blood cell production during homeostasis[62] than with other studies suggesting that HSCs contribute only during physiological stress conditions such as systemic infection or inflammation[63].

A recent study demonstrated that Tregs require CXCR4 for homing to the bone marrow where they play a direct role in controlling HSC homeostasis by secreting adenosine[24]. However, we failed to measure significant changes in the HSC compartment size when T cells lack CXCR4, including Tregs. A major difference between these studies is that Hirata et al. analyzed mice in which only Tregs are *Cxcr4*-deficient whereas in our studies we deleted *Cxcr4* in all T cell subsets. Under homeostasis, naïve T cells express very little CXCR4 and are largely unable to migrate into the bone marrow[64]. In contrast, memory T cells upregulate CXCR4 and migrate into the bone marrow where these cells receive homeostatic signals (presumably IL7 and/or IL15) required for their long-term maintenance[65,66]. Thus, it is possible that

adenosine produced by bone marrow Tregs suppresses bystander activation of recirculating memory T cells to reduce or prevent the release of inflammatory signals in the HSC niche[67].

In summary, our studies revealed that mSCF acts as a carrying capacity factor for HSC homeostasis due to competition from cKit-dependent hematopoietic progenitors for limited amounts of mSCF under homeostatic conditions. Furthermore, while MSPCs respond to long-range cues such as hormones for regulation and differentiation, these cells and ECs are impervious to alterations in cell-cell interactions with hematopoietic progenitors and remain transcriptionally stable.

# Methods

All research described in this study complies with ethical regulations according to the protocol approved by the Yale University Institutional Animal Care and Use Committee.

## Mice

C57BL/6NCR (strain code 556) (CD45.2+) and B6-Ly5.1/Cr (stain code 564) (CD45.1+) were purchased from Charles River Laboratories. *Rosa26mTmG*, *Pf4*-cre, *Kitl*[fl/fl], *Lepr*-cre, and *Kitl*[GFP/+] mice were from The Jackson Laboratories. *Cxcr4*[fl/fl], *Lyz2*-cre, *Il7ra*-cre, *CD4*-cre, and *Rosa26*[tdtomato/+] mice were from internal colonies. *Flk2*-cre mice were a gift from Dr. E. Camilla Forsberg (University of California, Santa Cruz). Although *Flk2*-cre transgene is inserted into Y-chromosome, our previous work showed the hematopoietic cell composition in bone marrow and secondary lymphoid organs of *Flk2*-cre transgenic mice is indistinguishable from their littermate controls. All mice analyzed were 8-20 week old, were maintained under specific pathogen-free conditions at Yale Animal Resources Center.

## In vivo treatments

Mice were treated with 5-FU at a dose of 150 mg/Kg/week and monitored closely for general condition. In vivo blocking of cKit signaling was achieved by treatment with anti-cKit blocking antibody (clone ACK2, no azide, low endotoxin, Biolegend) at a dose of 200 μg/mouse intravenously. In vivo BrdU incorporation studies were performed by injecting BrdU (BD Biosciences) dissolved in saline (1 mg/200 μL) intravenously.

## Flow cytometry

For analyses of hematopoietic cell composition, bone marrow cells were obtained by crushing long bones with DMEM supplemented with 2% FBS, 1% Penicillin/Streptomycin, 1% L-glutamine, and 1% HEPES. Spleen cells were obtained by mashing spleens through 70 μm cell strainers with the same media. Bone marrow stromal cell isolation and mSCF staining were performed as previously described[10]. Cells were counted with a Beckman Coulter Counter. Cells were then stained with antibody cocktail diluted in FACS buffer, at the concentration of 25 μl per 1 × 10⁶ cells on ice. To stain intracellular proteins and markers including BrdU, Ki67, pSTAT3, and pSTAT5, cells were fixed in Cytofix/Cytoperm solution (BD) for 20 min at a concentration of 25 μl per 1 × 10⁶ cells on ice and washed twice with PermWash buffer (BD). Cells were then permeabilized in Permeabilization Buffer Plus (BD) at the same concentration on ice for 10 min and washed twice. The Ki67 antibody was diluted in FACS buffer and used to stain cells on ice. The BrdU, pSTAT3 and pSTAT5 antibodies were diluted in PermWash buffer, and the cells were stained at room temperature. Hematopoietic cell populations were identified as follows: LSK: Lineage⁻ cKit⁺ SCA-1⁺; LT-HSC: Lineage⁻ cKit⁺ SCA-1⁺ FLT3⁻ CD150 (SLAM)⁺; ST-HSC: Lineage⁻ cKit⁺ SCA-1⁺ FLT3⁻ CD150⁻; MPP: Lineage⁻ cKit⁺ SCA-1⁺ FLT3⁺ CD150⁻ (The lineage cocktail was: CD19, B220, CD3e, CD4, Gr1, NK1.1, Ter119, CD11b, CD11c, CD41, CD48). MPP2: Lineage⁻ cKit⁺ SCA-1⁺ FLT3⁻ CD48⁺ SLAM⁺; MPP3: Lineage⁻ cKit⁺ SCA-1⁺ FLT3⁻ CD48⁺ SLAM⁻; MPP4: Lineage⁻ cKit⁺ SCA-1⁺ FLT3⁺ (The lineage cocktail was: CD19, B220, CD3e, CD4, Gr1,

NK1.1, Ter119, CD11b, CD11c). Treg: CD3e⁺ CD4⁺ CD25⁺; MSPC: CD45⁻ Ter119⁻ CD31⁻ CD144⁻ LEPR⁺; EC: CD45⁻ Ter119⁻ CD31[high] CD144[high]. Antibody fluorochromes, concentrations, and dilutions used are included in Supplementary Data 3.

## Immunostaining and Microscopy analyses

Freshly dissected femurs were fixed in 2% paraformaldehyde-based fixative in PBS at 4 °C overnight. Bones were dehydrated in a solution of 30% sucrose in PBS, at 4 °C overnight. Samples were embedded in OCT and snap-frozen in an ethanol/dry ice bath. Frozen sections were prepared according to the Kawamoto method or using the CryoJane tape transfer system (Leica). Femur whole mounts frozen in OCT were stained with primary antibodies for 2–3 days at 4 °C and secondary antibodies for 1 day at 4 °C. Slides were mounted with Fluormount-G (SourthernBiotech) or with a 30% glycerin solution. Images were acquired on a Leica SP8 confocal microscope.

## Competitive reconstitution assay

Recipient mice (CD45.1⁺) were exposed to whole-body lethal irradiation from a ¹³⁷Cs source (two doses of 550 rads separated by 3 h). Mice were then reconstituted with 5 × 10⁶ donor whole bone marrow cells (a mixture of CD45.2⁺ and CD45.1⁺ cells in a 1:1 ratio) by intravenous injection. Chimeric mice were analyzed 16 weeks after reconstitution. For long-term reconstitution analysis, 6 × 10⁶ donor whole bone marrow cells were injected to recipients, and the chimerism was analyzed 16 weeks after reconstitution in both primary and secondary transplantations.

## Bone marrow stromal cells preparation for bulk RNA sequencing

Long bones were flushed with HBSS supplemented with 2% of FBS, 1% Penicillin/Streptomycin, 1% L-glutamine, 1% HEPES, and 200 U/mL Collagenase IV (Worthington Biochemical Corporation). Cells were first digested at 37 °C for 20 min and gently pipetted to dissociate cell clumps. Cells were then filtered through 100 μm cell strainers after being digested at 37 °C for another 10 min, and washed with media. Stromal cells were enriched by depleting hematopoietic cells with biotin-conjugated CD45 and Ter119 antibodies, and Dynabeads® Biotin Binder (Invitrogen #11047). Cells were stained with CD31, CD144 and PDGFRα antibodies after depletion and CD45⁻ Ter119⁻ CD31⁻ CD144⁻ PDGFRα⁺ SCF-GFP⁺ cells were sorted into DMEM with 10% FBS. Sorting was performed by the BD FACS Aria II. The sorted cells were then sorted with the same gating strategy again to reduce contamination into 350 μL RLT plus buffer with 3.5 μL β-mercaptoethanol. RNA was extracted using the RNeasy® Plus Micro Kit (Qiagen #74034).

## Bulk RNA sequencing

RNA sequencing was performed using the Illumina HiSeq2500 system with paired-end 2 × 76 bp read length by the Yale Center for Genome Analysis. The sequencing reads were trimmed by 7 bp on the 5′-end and until QS ≥ 20 on the 3′-end. The sequencing reads were then aligned onto the reference genome Mus musculus GRCm38 (mm10) using HISAT2[68] and converted to BAM files using SAMtools[69]. Read counts were generated using HTSeq-count[70] with GENCODE v27 as the gene model. By utilizing DESeq2[71], the read counts were normalized by size factor and the differential expression of genes were calculated using LRT model in correction for batch effect.

## Bone marrow stromal cells preparation for single-cell RNA sequencing

Bone marrow stromal cells were isolated using the same method as the bulk RNA sequencing cell preparation. The bones after flushing were chopped into small pieces and digested with HBSS supplemented with 2% of FBS, 1% Penicillin/Streptomycin, 1% L-glutamine, 1% HEPES, and 200 U/mL Collagenase IV (Worthington Biochemical Corporation) at

37 °C for 45 min under agitation (120 rpm). Cells were then filtered through 100 μm cell strainers before combined with digested bone marrow stromal cells. Cells were then stained with CD31, CD144, lineage (B220, CD19, CD11b, Gr1, CD3e) and CD71 antibodies. CD45⁻ Ter119⁻ Lin⁻ CD31⁻ CD144⁻ CD71⁻ cells were sorted by the BD FACS Aria II machine into 350 μL DMEM with 20% FBS. Single-cell RNA sequencing was performed by the Yale Center for Genome Analysis. The libraries were prepared using the Chromium Single Cell 3′ Reagent Kits v3 according to the protocol. Libraries were run on an Illumina Nova-Seq system with 100-bp paired-end reads to ~80% saturation level and to get the coverage to ~40,000 reads per cell, and the sequencing reads were aligned onto Mus musculus GRCm38 (mm10) reference genome.

### scRNA-seq data preprocessing and analysis
Barcode processing and single cell 3′ gene count matrix calculation were conducted using the 10x Genomic Cell Ranger 4.0.0[72] and Gene-Barcode matrix containing 14,027 cells were generated. Quality control, finding highly variable genes, dimensionality reduction, graph-based unsupervised clustering, and identification of differentially expressed genes were performed using Seurat R Package 4.0[73]. Cells having fewer than 200 or more than 6000 detected features and more than 20% mitochondrial gene mapped reads were excluded from downstream analyses. Also, mitochondrial and ribosomal protein features were removed from the count matrix and contaminating hematopoietic cells expressing the genes listed in Supplementary Data 2 were filtered out. Then the count data was normalized, and variance stabilized using the SCTransform[74].

### Dimensionality reduction, unsupervised cell clustering, and visualization
With the normalized gene-barcode matrix of resulting 3818 cells, highly variable genes were detected and dimensionality reduction with principal component analysis (PCA) was conducted. The number of optimal principal components (PCs) was determined by *ElbowPlot* function of Seurat. Dimensionality was reduced and projected the cells in 2D space using UMAP by *RunUMAP* function of Seurat. Markov affinity-based graph imputation of cells[75] was used to denoising the high-dimensional scRNA-seq data by imputing plausible gene expression in each cell.

### Annotation of cell types and identification of cluster markers
The markers defining each cluster were identified by performing *FindAllMarkers* in the Seurat R package using the MAST method[76]. Feature plots with the top 10 significant positive markers of each cluster and a heatmap of the top 10 significant positive markers were generated to visualize how well the clusters are defined. Also, DEGs between conditions within the same cluster were identified using *FindMarkers* with the MAST method.

### Differential abundance test
Differential abundance of neighborhoods was analyzed via MiloR R package (https://github.com/MarioniLab/miloR) by allocating cells to partially overlapping neighborhoods on a k-nearest neighbor (KNN) graph. With calculated log2 fold change and FDR, clusters having statistically significant differential abundance were determined.

### Elisa assays
To harvest bone marrow interstitial fluid, long bones were flushed into 1.5 ml Eppendorf tubes using compressed air and immediately weighed and dissociated in 15-fold DPBS by pipetting and vortexing. The mixture was then centrifuged at 100 g for 5 min, and the supernatant was transferred to new tubes, followed by another centrifugation at 9200 g for 5 min to remove the remaining debris. To harvest

serum, blood was collected and allowed to clot at room temperature for 1 h. The clotted blood was then centrifuged at 150 g for 10 min and the serum was transferred to new tubes. The serum was diluted 5-fold for the assay. ELISA for SCF and TPO was conducted with Mouse SCF Quantikine ELISA Kit (R&D, MCK00) and Mouse Thrombopoietin Quantikine ELISA Kit (R&D, MTP00).

### Reporting summary
Further information on research design is available in the Nature Research Reporting Summary linked to this article.

## Data availability
Source data are provided with this paper. All bulk RNA-seq and scRNA-seq data were deposited in the Gene Expression Omnibus (GEO) under the accession number GSE171015. Source data are provided with this paper.

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

## Acknowledgements

We thank Vivian Y. Lim, Abigail Jarret, and Tianyang Mao (Yale University) for help with some experiments; Aaron Ring and Shangqin Guo (Yale University) for insightful suggestions and technical support; and Sean J. Morrison and Stefano Comazzetto (UT Southwestern) for sharing biological samples. These studies were funded by the NIH (RO1AI113040 and R21AI13306001A1 to J.P. Pereira), and by the National Research Foundation of Korea (NRF) grant funded by the Korea government (MSIT) (No. 2020R1F1A1076705).

## Author contributions

R.M. contributed to conceptualizing this project, experimental design, executed most experiments, and helped writing the manuscript. H.C. and J.C. analyzed all RNA sequencing datasets and contributed to writing the manuscript. X.F. and A.C.G. executed some experiments and contributed to writing the manuscript. J.P.P. conceptualized and supervised project, contributed to data analyses, and wrote the manuscript

## Competing interests

The authors declare no competing interests.
