## [Peer Review File · Nature Communications]

Competition between hematopoietic stem and progenitor cells controls hematopoietic stem cell compartment sizeREVIEWER COMMENTS

Reviewer #1 (Remarks to the Author):

Miao et al. provide evidence that competition between HSCs and MPPs/restricted progenitors for SCF produced by LepR⁺ cells limits HSC frequency in the bone marrow. The evidence for this came from an elegant series of experiments in which they depleted MPPs and restricted progenitors by deleting CCR4 from those cells, rendering them insensitive to the niche factor Cxcl12. This led to increased SCF expression by LepR⁺ cells and increased HSC frequency. There are a number of appealing aspects of this paper. One of the most important and understudied aspects of niche biology is the characterization of feedback mechanisms that regulate HSC and progenitor frequency. This paper offers a plausible mechanism by which HSCs expand in number after the depletion of MPPs/restricted progenitors.

1. The authors have taken rigorous approaches to test a novel hypothesis but there remains an alternative interpretation. We know that anything that depletes restricted progenitors or differentiated cells causes the activation of HSCs, and often extramedullary hematopoiesis, to regenerate the missing hematopoietic cells. That's essentially what the authors see and they propose that it's caused by decreased competition for SCF expressed by LepR⁺ cells; however, it remains possible that there is a distinct mechanism, independent of SCF, that promoted HSC expansion after the depletion of MPPs/restricted progenitors. If so, SCF competition may not be the primary quorum-sensing mechanism. The authors go to impressive lengths to exclude alternative possibilities, even doing single cell RNAseq on the stromal cells to provide evidence they are not upregulating the expression of other HSC niche factors. With the exception of some concerns highlighted below, it would probably be impossible for the authors to completely exclude alternative possibilities. Nonetheless, the authors should acknowledge the possibility of SCF-independent mechanisms.
2. The authors conclude that membrane bound SCF levels on LepR⁺ cells increase as a result of "reduced consumption" by ckit⁺ progenitors. However, they offer no evidence that those progenitors are "consuming" SCF. They show only that when progenitors are depleted, mSCF levels go up. There are many possibilities for why SCF levels go up, including more indirect quorum sensing mechanisms that increase SCF expression whenever any kind of a progenitor is depleted. The idea that progenitors are directly consuming SCF made by stromal cells is an over-interpretation that should be removed.
3. It's impressive that the authors made the Cxcr4/SCF double knockout mice to test whether heterozygosity for SCF could rescue the Cxcr4 phenotype. However, they didn't show enough data from these experiments to conclude that SCF heterozygosity was actually rescuing the Cxcr4 phenotype (Suppl. Fig. 4E). Instead of just showing HSC numbers in the Cxcr4 mutant and double mutant mice, they also have to show HSC numbers in the wild-type control mice and SCF heterozygous mice, side-by-side, from the same experiment. SCF heterozygosity by itself reduces HSC frequency. To conclude that it's actually rescuing the Cxcr4 phenotype, it would have to reduce HSC frequency to a greater extent in the Cxcr4 deficient background as compared to the Cxcr4 wild-type background. If that's not the case, then the authors should tone down the conclusion that the increase in HSC frequency after Cxcr4 deletion is caused by the increase in SCF.
4. In the second section of the results (line 141) the authors conclude that "HSC quiescence and self-renewal are independent of CXCR4-mediated hematopoietic progenitor localization". This is over-interpreted. First, there are no data on progenitor localization so I'd avoid that word. Second, although the authors conclude there is no loss of HSC quiescence they show a significant increase in the number of dividing HSCs. They should avoid this

disconnect between the data and the conclusions. I understand that the authors want to argue that the absolute number of non-dividing HSCs remains higher after Cxcr4 deletion than in control mice, despite the increase in the number of dividing HSCs. I'm fine with that, but they shouldn't claim there is no change in quiescence because many HSCs do go into cycle after progenitor depletion, as would be expected.

5. It's impressive that the authors also deleted Cxcr4 from the megakaryocytes, macrophages, lymphoid cells, or T cells (lines 171-190). It would be helpful if they could name the Cre alleles they used in the main text. They also should show overall bone marrow and spleen cellularity, and blood cell counts, for each of these genetic backgrounds to make clear whether there are gross hematopoietic defects in any of these mice.

6. The authors conclude in Figure 5 that Cxcr4 deletion from LepR+ cells did not affect the expression of "hematopoietic cytokines, chemokines, nor genes known to directly or indirectly regulate HSC homeostasis (Fig. 5D)". This doesn't seem consistent with the data in Figure 5D, which show changes in the expression of Cxcl9, Ccl9, Igfbp2, and Igf2bp2. It seems like that statement should be softened.

7. Was the ACK2 antibody used in Figure 6 to deplete ckit+ cells confirmed to be endotoxin free? If not, the effects on SCF expression could have been caused by inflammation/endotoxin rather than progenitor depletion.

8. In Figure 1F, the authors should show the levels of multilineage reconstitution by donor myeloid, B, and T cells in the blood over time, not just donor HSC frequency.

Reviewer #2 (Remarks to the Author):

Miao et al. described that HSC compartment size was regulated by cell-cell competition between HSCs and progenitors for niche factors produced by MSPCs and ECs. Conditional knockout of CXCR4 in MPPs resulted in HSC expansion without any major changes in HSC quiescence and self-renewal nor in MSPC and EC transcriptome. Mechanistically, they found that mSCF on niche cells could control HSC compartment size due to their consumption competition between HSCs and hematopoietic progenitors. These findings suggest an active model of niche regulation in HSC compartment during homeostasis through cell competition.

Major concerns:

1. Although the authors performed in vitro and in vivo experiments to verify mSCF expression, in situ immunofluorescence analysis could be helpful to compare the different expression of CXCR4 on HSCs and progenitor cells, and mSCF on niche cells.
2. The results showed that the increased HSCs in CXCR4 cKOs were due to the increased mSCF availability on niche cells. They also detected a significant increase of mSCF on niche cells without any obvious changes at transcription level. So, why and how did CXCR4 cKO in MPPs lead to increased mSCF on MSPCs and ECs? And which is the exact reason for increased HSCs, i.e. increased availability or increased expression of mSCF?
3. Did CXCR4 deletion in MPPs block HSC differentiation, then induce HSC expansion?
4. Why was the protein level of mSCF increased in Cxcr4 cKO mice? how about the protein level of mSCF and HSC number in KitlGFP/+ mice? and the potential relationship and regulatory role between Cxcr4 and SCF are unclear.
5. Given that SCF is one of important cytokines in HSC niche, whether the cellular

competition between HSC and MPPs is mSCF-specific, or other growth factors also have similar function, need to be addressed or at least discussed.

6. The downstream mechanism about JAK/STAT pathway should be further verified.

Minor concerns:

1. The Results part should be rewritten, as the current version contains lots of methods details.

Reviewer #3 (Remarks to the Author):

The chemokine CXCL12 and its receptor CXCR4 are essential for the maintenance of HSCs and production of immune cells in the bone marrow. The authors have demonstrated that mice lacking CXCR4 in flt2+ multipotent hematopoietic progenitors and their progeny (Flk2-cre; CXCR4fl/fl mice), HSC numbers were increased by 2-fold although their reconstitution potential and CXCR4 expression were normal. Additionally, they found that protein expression of membrane-bound stem cell factor (mSCF) on Leptin receptor+ mesenchymal stem/progenitor cells (MSPCs) and endothelial cells (ECs) was increased in Flk2-cre; CXCR4fl/fl mice as well as Ack2 (anti-c-kit antibodies)-treated wild-type mice, in which c-kit+ cells were severely reduced. In contrast to Flk2-cre; CXCR4fl/fl mice, HSC numbers were unaltered in mice lacking CXCR4 in myeloid, lymphoid, or megakaryocyte lineage cells. Furthermore, the reduction of SCF by half in Flk2-cre; CXCR4fl/fl mice decreased HSC numbers and cell-surface expression of mSCF on MSPCs to normal levels. Based on these findings, the authors suggest that HSCs and proximally located hematopoietic progenitors are regulated by cell competition for limited amounts of mSCF. These results are very interesting and open new views on the regulation of the numbers of HSCs and hematopoietic progenitors in hematopoiesis. However, there are several concerns with the conclusions that can be drawn at this stage.

1. It has been shown previously that HSC numbers were normal in SCF+/- (heterozygous) mice (Comazzeto et al., Cell Stem Cell 24;1,2019), supporting the authors' idea that a reduction in mSCF expression on MSPCs reduced the HSC numbers in SCF gfp/+ (heterozygous) Flk2-cre; CXCR4fl/fl mice compared with Flk2-cre; CXCR4fl/fl mice. Thus, I would recommend the authors to show HSC numbers in SCF gfp/+ mice in Figure 6F to strengthen their proposition.

2. The population of c-kit+ cells which was reduced in number in Flk2-cre; CXCR4fl/fl mice is the Lin-c-kit+Sca-1+flt2+ MPP4, suggesting that MPP4 cells reduced mSCF expression on MSPCs. However, a HSC is not in contact with a MPP4 cell or the MSPC which a MPP4 cell contact (Figure 7D in the study by Cordeiro Gomes et al., Immunity 2016)), arguing against the authors' conclusion that HSC numbers are regulated by MPP4 cells by cell competition for limited amounts of mSCF. Thus, another mechanistic experiments and insight would be needed to explain the increased HSC numbers in Flk2-cre; CXCR4fl/fl mice.

3. Along similar lines, the numbers of HSCs and MPP4 cells are too small compared with those of MSPCs to affect mSCF expression on the majority of MSPCs. On the other hand, the numbers of neutrophils and mature B cells were reduced in Flk2-cre; CXCR4fl/fl mice (Supplementary Figure 1B) and the numbers of myeloid cells but not B cells have been shown to be severely reduced in Ack2-treated wild-type mice (Ogawa et al., JEM 174; 63,

1991). These results raise the possibility that neutrophils provide the signals that reduce the mSCF expression on the majority of MSPCs. The authors should show the numbers of neutrophils in Ack2-treated wild-type mice and address this possibility.

4. The information of ages of analyzed mice should be provided. How about HSC numbers and mSCF expression on MSPCs and ECs in juvenile and aged Flk2-cre; CXCR4fl/fl mice?

Minor points

1. Why are there differences in HSC numbers in control bone marrow between Figure 1 and Supplementary Figure 1C?

2. Why are there differences in MPP4 numbers in control bone marrow between Figure 1 and Supplementary Figure 1C?

3. Line 238, what is the evidence that Ptn (pleiotrophin) or Icam1 is essential for HSC behavior?

4. Line 238, S6C should be S4C.

We were very pleased that the three reviewers considered these studies to be insightful and interesting, with only relatively minor questions or concerns being raised. We believe that we addressed all of the questions and concerns raised by the reviewers and in that process we made significant improvements to the original manuscript. Thus, we are grateful to the time spent by the reviewers in examining our studies. Below is a point-by-point response to all of the reviewers comments and concerns. The revised manuscript includes changes highlighted in **yellow**.

Reviewer #1 (Remarks to the Author):

Miao et al. provide evidence that competition between HSCs and MPPs/restricted progenitors for SCF produced by LepR+ cells limits HSC frequency in the bone marrow. The evidence for this came from an elegant series of experiments in which they depleted MPPs and restricted progenitors by deleting CCR4 from those cells, rendering them insensitive to the niche factor Cxcl12. This led to increased SCF expression by LepR+ cells and increased HSC frequency. There are a number of appealing aspects of this paper. One of the most important and understudied aspects of niche biology is the characterization of feedback mechanisms that regulate HSC and progenitor frequency. This paper offers a plausible mechanism by which HSCs expand in number after the depletion of MPPs/restricted progenitors.

1. The authors have taken rigorous approaches to test a novel hypothesis but there remains an alternative interpretation. We know that anything that depletes restricted progenitors or differentiated cells causes the activation of HSCs, and often extramedullary hematopoiesis, to regenerate the missing hematopoietic cells. That's essentially what the authors see and they propose that it's caused by decreased competition for SCF expressed by LepR+ cells; however, it remains possible that there is a distinct mechanism, independent of SCF, that promoted HSC expansion after the depletion of MPPs/restricted progenitors. If so, SCF competition may not be the primary quorum-sensing mechanism. The authors go to impressive lengths to exclude alternative possibilities, even doing single cell RNAseq on the stromal cells to provide evidence they are not upregulating the expression of other HSC niche factors. With the exception of some concerns highlighted below, it would probably be impossible for the authors to completely exclude alternative possibilities. Nonetheless, the authors should acknowledge the possibility of SCF-independent mechanisms.

We agree with the reviewer comment and have acknowledged the possibility that SCF-independent mechanisms contributed to the HSC expansion described. This discussion is included in the Discussion section.

2. The authors conclude that membrane bound SCF levels on LepR+ cells increase as a result of "reduced consumption" by ckit+ progenitors. However, they offer no evidence that those progenitors are "consuming" SCF. They show only that when progenitors are depleted, mSCF levels go up. There are many possibilities for why SCF levels go up, including more indirect quorum sensing mechanisms that increase SCF expression whenever any kind of a progenitor is depleted. The idea that progenitors are directly consuming SCF made by stromal cells is an over-interpretation that should be removed.

We understand the reviewer concern. Although prior studies have pinpointed the exact cellular sources of SCF for HSCs and for downstream hematopoietic progenitors (Ding Nature 2012, 2013; Greenbaum Nature 2013; Comazzeto et al., Cell Stem Cell 2019), whether progenitors are actively (continuously) consuming SCF wasn't entirely clear. We addressed this question by performing a short-term cKit blocking experiment in vivo (24h) combined with BrdU incorporation. We demonstrate that multiple hematopoietic progenitors are significantly affected by short-term cKit blocking in vivo, and these data are included in a revised Fig. S5.

We also attempted to determine how hematopoietic progenitors consume mSCF from MSPCs. We postulated HSCs and hematopoietic progenitors may "nibble" on MSPCs as these cells physically interact in the BM, and remove membrane fragments, a process known as trogocytosis. To test this hypothesis we prepared a set of BM chimeras in which hematopoietic cells are non-fluorescent, MSPC and MSPC-derived cells express membrane-bound GFP, and all other stromal/endothelial cells express membrane-bound Tomato fluorescent proteins. As controls, we used non-fluorescent hematopoietic cells developing in a non-fluorescent stromal/endothelial cell environment using congenic markers (CD45.1 and CD45.2). Specifically, we crossed LeprCre/+ mice with Rosa26 mT/mG mice (Muzumdar et al. Genesis 2007); these mice were lethally irradiated and reconstituted with non-fluorescent C57BL6/J (CD45.1) bone marrow cells. Control experiments used C57BL6/J mice (CD45.1) as non fluorescent recipients and C57BL6/J mice (CD45.2) as BM cell donors. By measuring mGFP fluorescence on hematopoietic cells isolated from LeprCre/+ Rosa26 mT/mG (and compare to background fluorescence in cells isolated from C57BL6/J chimeras) we were able to detect highest mGFP intensity on HSCs and on several hematopoietic progenitors. In contrast, phagocytic monocytes did not show measurable mGFP fluorescence, and even neutrophils showed minimal acquisition of mGFP. These data were included as a new supplementary figure 6A-C. To exclude the possibility that hematopoietic progenitors and differentiated immune cells acquired mGFP ex-vivo during tissue processing (i.e. during BM flushing, single cell suspension and incubations for FACS staining, etc) we performed parallel experiments in which we mixed non-fluorescent (CD45.2+) C57BL6/J BM cells with BM cells isolated from LeprCre/+ Rosa26 mT/mG full chimeras (CD45.1+). This experiment shows minimal acquisition of mGFP fluorescence in hematopoietic progenitors ex-vivo (Sup Figure 6D).

3. It's impressive that the authors made the Cxcr4/SCF double knockout mice to test whether heterozygosity for SCF could rescue the Cxcr4 phenotype. However, they didn't show enough data from these experiments to conclude that SCF heterozygosity was actually rescuing the Cxcr4 phenotype (Suppl. Fig. 4E). Instead of just showing HSC numbers in the Cxcr4 mutant and double mutant mice, they also have to show HSC numbers in the wild-type control mice and SCF heterozygous mice, side-by-side, from the same experiment. SCF heterozygosity by itself reduces HSC frequency. To conclude that it's actually rescuing the Cxcr4 phenotype, it would have to reduce HSC frequency to a greater extent in the Cxcr4 deficient background as compared to the Cxcr4 wild-type background. If that's not the case, then the authors should tone down the conclusion that the increase in HSC frequency after Cxcr4 deletion is caused by the increase in SCF.

We appreciate and agree with the reviewer concern. We did not include data from SCF WT and SCF Het mice because these data have been previously published by others, as also pointed out

by reviewer 3. In the revised manuscript we now include HSC numbers in SCF WT and SCF Het mice that are also hemizygous for *Cxcr4* in MPPs and downstream hematopoietic cells (revised Figure 6E and F). We show that *Scf* heterozygosity only brings down HSC numbers in *Cxcr4cKO* mice, which further validates our conclusion that reducing *Scf* availability rescued the HSC expansion seen in *Cxcr4cKO* mice.

4. In the second section of the results (line 141) the authors conclude that “HSC quiescence and self-renewal are independent of CXCR4-mediated hematopoietic progenitor localization”. This is over-interpreted. First, there are no data on progenitor localization so I’d avoid that word. Second, although the authors conclude there is no loss of HSC quiescence they show a significant increase in the number of dividing HSCs. They should avoid this disconnect between the data and the conclusions. I understand that the authors want to argue that the absolute number of non-dividing HSCs remains higher after *Cxcr4* deletion than in control mice, despite the increase in the number of dividing HSCs. I’m fine with that, but they shouldn’t claim there is no change in quiescence because many HSCs do go into cycle after progenitor depletion, as would be expected.

We agree with the reviewer comment and revised the text to more accurately describe our experiments. The text was changed to “CXCR4 deletion from hematopoietic progenitor cells and effects on HSC quiescence and self-renewal”

5. It’s impressive that the authors also deleted *Cxcr4* from the megakaryocytes, macrophages, lymphoid cells, or T cells (lines 171-190). It would be helpful if they could name the Cre alleles they used in the main text. They also should show overall bone marrow and spleen cellularity, and blood cell counts, for each of these genetic backgrounds to make clear whether there are gross hematopoietic defects in any of these mice.

This information has been added to the Results section.

6. The authors conclude in Figure 5 that *Cxcr4* deletion from *LepR+* cells did not affect the expression of “hematopoietic cytokines, chemokines, nor genes known to directly or indirectly regulate HSC homeostasis (Fig. 5D)”. This doesn’t seem consistent with the data in Figure 5D, which show changes in the expression of *Cxcl9*, *Ccl9*, *Igfbp2*, and *Igf2bp2*. It seems like that statement should be softened.

We appreciate the reviewer comment and revised the text to not only soften this conclusion but also to bring forward the possibility that *Igfbp2* and *Igf2bp2* increased expression may contribute to increasing HSC numbers in cKO mice.

7. Was the ACK2 antibody used in Figure 6 to deplete *ckit+* cells confirmed to be endotoxin free? If not, the effects on SCF expression could have been caused by inflammation/endotoxin rather than progenitor depletion.

The ACK2 was confirmed to be azide free and undetectable endotoxin. Having said this, the effects of inflammation on SCF expression have been studied by others (Ueda et al JEM 2004 and 2005) and have shown significant decrease in mRNA and protein levels after systemic inflammation. In contrast, we find increased SCF levels in ACK2-treated mice

8. In Figure 1F, the authors should show the levels of multilineage reconstitution by donor myeloid, B, and T cells in the blood over time, not just donor HSC frequency.

As we have described in a prior study (Cordeiro Gomes et al Immunity 2016), CXCR4 is required in hematopoietic progenitors for their proper differentiation into lymphoid and myeloid cell lineages. Therefore, the level of hematopoietic reconstitution in lymphoid and myeloid lineages in cKO mixed chimeras does not reflect the actual reconstitution efficiency. Because of these well described effects of CXCR4 in hematopoiesis we consider that including these data in the main text will not benefit the reader when examining this study.

Reviewer #2 (Remarks to the Author):

Miao et al. described that HSC compartment size was regulated by cell-cell competition between HSCs and progenitors for niche factors produced by MSPCs and ECs. Conditional knockout of CXCR4 in MPPs resulted in HSC expansion without any major changes in HSC quiescence and self-renewal nor in MSPC and EC transcriptome. Mechanistically, they found that mSCF on niche cells could control HSC compartment size due to their consumption competition between HSCs and hematopoietic progenitors. These findings suggest an active model of niche regulation in HSC compartment during homeostasis through cell competition.

Major concerns:

1. Although the authors performed in vitro and in vivo experiments to verify mSCF expression, in situ immunofluorescence analysis could be helpful to compare the different expression of CXCR4 on HSCs and progenitor cells, and mSCF on niche cells.

We respectfully disagree with the reviewer on these two points. First, to our knowledge there is no study that has reliably demonstrated SCF protein expression on bone marrow niche cells by in situ immunofluorescence. As it is clear from the data we presented in this study, a reliable anti-SCF antibody fails to detect membrane-bound SCF in wild type mice by flow cytometry (Sup Fig. 4C). Given the higher sensitivity of flow cytometer detectors relative to confocal microscopy signal detection we are certain that in situ mSCF stains will not allow visualization of mSCF on niche cells. We remind the reviewer that isotype control staining are inappropriate negative controls given that different antibodies behave very differently in antigen cross-reactivity. Regarding CXCR4, given that this receptor is expressed in > 95% of bone marrow leukocytes we do not understand how in situ CXCR4 staining would benefit this study.

2. The results showed that the increased HSCs in CXCR4 cKOs were due to the increased mSCF availability on niche cells. They also detected a significant increase of mSCF on niche cells without any obvious changes at transcription level. So, why and how did CXCR4 cKO in MPPs lead to increased mSCF on MSPCs and ECs? And which is the exact reason for increased HSCs, i.e. increased availability or increased expression of mSCF?

We thank the reviewer for raising this question as we have spent a considerable amount of time and resources attempting to address it. When CXCR4 is deleted in MPPs using Flk2-cre essentially all downstream hematopoietic progenitors and differentiated hematopoietic cells will also be CXCR4 deficient. This is a large fraction of all hematopoietic cells in the bone marrow, of which a significant fraction expresses the SCF receptor cKit. In the revised manuscript, we included data showing that cKit⁺ hematopoietic progenitors are continuously consuming SCF (see new Fig. S6). Thus, the simplest explanation of SCF increase in Cxcr4 cKO mice is that cKit⁺ hematopoietic progenitors require CXCR4 for consuming mSCF displayed on bone marrow MSPCs. When CXCR4 is deleted from MPPs and MPP-differentiated cells, these cells become unable to consume SCF, which results in its increased availability. In turn, our data indicates strongly that increased SCF availability leads to an increase in the total number of quiescent and non-quiescent HSCs, thus revealing a new layer of regulation of the HSC compartment size. We have revised the discussion of the manuscript to explain this rationale with more clarity.

3. Did CXCR4 deletion in MPPs block HSC differentiation, then induce HSC expansion?

Please see our arguments in point 2. Briefly, the simplest explanation is that CXCR4 deficiency reduces hematopoietic cell access to CXCL12-producing MSPCs and ECs. Because bone marrow niche cells express a constellation of hematopoietic cytokines that predominantly act in short-range manner, deficiency in CXCR4 impacts MPP and hematopoietic progenitor access to hematopoietic cytokines (see Miao et al 2020 *Frontiers in Immunol.* for a detailed overview of key hematopoietic cytokines expressed by CXCL12-producing niche cells and hematopoietic defects seen in CXCR4-deficient and CXCR4 conditionally deficient mice).

4. Why was the protein level of mSCF increased in Cxcr4 cKO mice? how about the protein level of mSCF and HSC number in KitlGFP/+ mice? and the potential relationship and regulatory role between Cxcr4 and SCF are unclear.

We thank the reviewer for raising this issue. We have now included data on mSCF levels and HSC numbers detected in KitlGFP/+ mice (revised Fig. 6E and 6F), which further supports our conclusion that increased mSCF levels result in increased numbers of HSCs.

5. Given that SCF is one of important cytokines in HSC niche, whether the cellular competition between HSC and MPPs is mSCF-specific, or other growth factors also have similar function, need to be addressed or at least discussed.

We thank the reviewer for raising this issue. We revised the discussion to incorporate this important issue.

6. The downstream mechanism about JAK/STAT pathway should be further verified.

Although we understand the reviewer interest in further studies on cKit/Jak/Stat pathway, we respectfully consider these studies to fall outside of the scope of this manuscript.

Minor concerns:

1. The Results part should be rewritten, as the current version contains lots of methods details.

We've shortened the results section to avoid excessive methodological details.

Reviewer #3 (Remarks to the Author):

The chemokine CXCL12 and its receptor CXCR4 are essential for the maintenance of HSCs and production of immune cells in the bone marrow. The authors have demonstrated that mice lacking CXCR4 in flt2+ multipotent hematopoietic progenitors and their progeny (Flk2-cre; CXCR4fl/fl mice), HSC numbers were increased by 2-fold although their reconstitution potential and CXCR4 expression were normal. Additionally, they found that protein expression of membrane-bound stem cell factor (mSCF) on Leptin receptor+ mesenchymal stem/progenitor cells (MSPCs) and endothelial cells (ECs) was increased in Flk2-cre; CXCR4fl/fl mice as well as Ack2 (anti-c-kit antibodies)-treated wild-type mice, in which c-kit+ cells were severely reduced. In contrast to Flk2-cre; CXCR4fl/fl mice, HSC numbers were unaltered in mice lacking CXCR4 in myeloid, lymphoid, or megakaryocyte lineage cells. Furthermore, the reduction of SCF by half in Flk2-cre; CXCR4fl/fl mice decreased HSC numbers and cell-surface expression of mSCF on MSPCs to normal levels. Based on these findings, the authors suggest that HSCs and proximally located hematopoietic progenitors are regulated by cell competition for limited amounts of mSCF. These results are very interesting and open new views on the regulation of the numbers of HSCs and hematopoietic progenitors in hematopoiesis. However, there are several concerns with the conclusions that can be drawn at this stage.

1. It has been shown previously that HSC numbers were normal in SCF+/- (heterozygous) mice (Comazzeto et al., Cell Stem Cell 24;1,2019), supporting the authors' idea that a reduction in mSCF expression on MSPCs reduced the HSC numbers in SCF gfp/+ (heterozygous) Flk2-cre; CXCR4fl/fl mice compared with Flk2-cre; CXCR4fl/fl mice. Thus, I would recommend the authors to show HSC numbers in SCF gfp/+ mice in Figure 6F to strengthen their proposition.

We thank the reviewer for the suggestion; this has been added to a revised Fig 6F.

2. The population of c-kit+ cells which was reduced in number in Flk2-cre; CXCR4fl/fl mice is the Lin-c-kit+Sca-1+flk2+ MPP4, suggesting that MPP4 cells reduced mSCF expression on MSPCs. However, a HSC is not in contact with a MPP4 cell or the MSPC which a MPP4 cell contact (Figure 7D in the study by Cordeiro Gomes et al.,(Immunity 2016)), arguing against the authors' conclusion that HSC numbers are regulated by MPP4 cells by cell competition for limited amounts of mSCF. Thus, another mechanistic experiments and insight would be needed to explain the increased HSC numbers in Flk2-cre; CXCR4fl/fl mice.

We understand the reviewer question but disagree with interpretation of our data. In Flk2-cre Cxcr4fl/fl mice all cKit+ progenitor cells are CXCR4 deficient and become numerically reduced in the bone marrow, as we have previously described in Cordeiro Gomes et al Immunity 2016. In the revised discussion of the manuscript we include a new paragraph explaining that cKit+ progenitors (which include MPP4, CMPs, GMPs, MEPs, MDPs, CLPs, and more differentiated but still cKit+ progenitors, such as proB cells, ILC precursors, NK cell precursors, etc) utilize CXCR4 for migration towards CXCL12-producing niche cells (predominantly MSPCs and ECs)

and presumably for encountering SCF produced by niche cells. In the absence of CXCR4, cKit+ progenitors reduce SCF consumption which in turn results in increased SCF availability. The reviewer points to our prior studies showing that HSCs localize in contact with niche cells mostly alone and only about 5% of HSCs being found in the vicinity of an MPP4. The reviewer argues that this infrequent co-localization argues against our proposed model of competition. However, the reviewer should consider the fact that HSCs are not entirely static (and so are MPPs, and other cKit+ progenitors) and can migrate significant distances over time. Thus, we argue that HSCs (and cKit+ progenitors) must interact with multiple MSPCs and ECs over time and receive signals provided by distinct niche cells. Because, in situ imaging of fixed tissue sections captures a single timepoint only, this type of data should not be over-interpreted.

3. Along similar lines, the numbers of HSCs and MPP4 cells are too small compared with those of MSPCs to affect mSCF expression on the majority of MSPCs. On the other hand, the numbers of neutrophils and mature B cells were reduced in Flk2-cre; CXCR4fl/fl mice (Supplementary Figure 1B) and the numbers of myeloid cells but not B cells have been shown to be severely reduced in Ack2-treated wild-type mice (Ogawa et al., JEM 174; 63, 1991). These results raise the possibility that neutrophils provide the signals that reduce the mSCF expression on the majority of MSPCs. The authors should show the numbers of neutrophils in Ack2-treated wild-type mice and address this possibility.

We thank the reviewer for bringing this important question. We have revised the Ack2 treatment data and found a small reduction in neutrophil numbers (~ 20%). The small reduction is consistent with the findings described in the Ogawa et al study, although the magnitude in neutrophil reduction seen in our study is considerably lower. We believe that differences in experimental design between these two studies underscores this difference. We treated mice for 3 days with 200ug, whereas Ogawa et al treated for 12 days with 1mg every two days. We've included these data and extended the discussion to raise the possibility that neutrophils may contribute to modulate mSCF availability.

4. The information of ages of analyzed mice should be provided. How about HSC numbers and mSCF expression on MSPCs and ECs in juvenile and aged Flk2-cre; CXCR4fl/fl mice?

This information has now been added in the Methods section. Unfortunately, we have not examined aged Flk2-cre; CXCR4fl/fl mice nor have access to 20-month old mice for generating these data. Given that the focus of these studies was on young adult/adult mice we consider that this request falls outside of the scope of this manuscript.

Minor points

1. Why are there differences in HSC numbers in control bone marrow between Figure 1 and Supplementary Figure 1C?

We apologize for this inconsistency. The major causes for this difference are the fact that mice in Fig 1C were a few weeks older (~ 16 weeks old) than in S1B (6-8 weeks), and the staining protocol was also different. The reason for the staining difference was the fact that we were interested in capturing MPP2, 3 and MPP4 populations described by the Passegue lab, which required using a different fluorochrome for lineage antibodies that had lower resolution. Overall,

it is the CTR vs KO comparison that is essential for analyses and thus we consider this small discrepancy to not affect the overall conclusions of this manuscript.

2. Why are there differences in MPP4 numbers in control bone marrow between Figure 1 and Supplementary Figure 1C?

We apologize for this inconsistency. The major causes for this difference are the fact that mice in Fig 1C were a few weeks older (~ 16 weeks old) than in S1B (6-8 weeks), and the staining protocol was also different. The reason for the staining difference was the fact that we were interested in capturing MPP2, 3 and MPP4 populations described by the Passegue lab, which required using a different fluorochrome for lineage antibodies that had lower resolution. Overall, it is the CTR vs KO comparison that is essential for analyses and thus we consider this small discrepancy to not affect the overall conclusions of this manuscript.

3. Line 238, what is the evidence that Ptn (pleiotrophin) or Icam1 is essential for HSC behavior?

References were included to support these statements.

4. Line 238, S6C should be S4C.

This has been corrected.

REVIEWER COMMENTS

Reviewer #1 (Remarks to the Author):

The authors have addressed my comments. I recommend the paper for publication.

Reviewer #2 (Remarks to the Author):

My previous concerns on the mSCF availability and its protein level in the knockout mice was still not well addressed, and the authors only provided some speculation without any direct evidence in the revision. The relative distribution of mSCF or CXCR4 was also not explored at all.

Reviewer #3 (Remarks to the Author):

The authors have given a satisfactory response to my concerns.

Reviewer #2 (Remarks to the Author):

We thank the reviewer for going through the revised manuscript and regret that it is still not acceptable for publication by this reviewer. The reviewer states that “My previous concerns on the mSCF availability and its protein level in the knockout mice was still not well addressed, and the authors only provided some speculation without any direct evidence in the revision. The relative distribution of mSCF or CXCR4 was also not explored at all.” These comments were solely based on a previous comment made by the reviewer that we also paste below:

1. Although the authors performed in vitro and in vivo experiments to verify mSCF expression, in situ immunofluorescence analysis could be helpful to compare the different expression of CXCR4 on HSCs and progenitor cells, and mSCF on niche cells.

As we stated before, to our knowledge, there is no study that has reliably demonstrated SCF protein expression on bone marrow niche cells by in situ immunofluorescence. As it is clear from the data we presented in this study, a reliable anti-SCF antibody fails to detect membrane-bound SCF in wild type mice by flow cytometry (Sup Fig. 4C). Given the higher sensitivity of flow cytometer detectors relative to confocal microscopy signal detection we are certain that in situ mSCF stains will not allow visualization of mSCF on niche cells. Besides these technical issues, we would like to express our agreement with this reviewer when he/she stated in the first round of reviews that “The results showed that the increased HSCs in CXCR4 cKOs were due to the increased mSCF availability on niche cells. They also detected a significant increase of mSCF on niche cells without any obvious changes at transcription level.” This is the fundamental new finding of our study that constitutes its main message. From this statement it is clear that the reviewer agrees with this main message. Like so many studies, new findings propel new questions and in our case, the exact mechanism that leads to increased mSCF continues to be defined. In the revised mechanism, we provided new and compelling data indicating that trogocytosis is a likely mechanism, a possibility that was well received by reviewers 1 and 3, and that seems to have been well received by reviewer 2 given the fact that no concerns were raised with those experiments or conclusions.

Regarding CXCR4, we conditionally deleted CXCR4 in >95% of hematopoietic cells in the bone marrow using a genetic strategy previously published by us (Gomes et al Immunity 2016) and by others (Forsberg and colleagues Cell Stem Cell 2011), and Cxcr4 deletion has been thoroughly confirmed in this mouse model. We continue to not understand in what way would performing IF analyses on CXCR4 expression would benefit this study. We are deeply concerned that adding more experiments without an underlying hypothesis weakens this study.

In conclusion,, for all the arguments presented above we strongly consider that the current manuscript contains all of the necessary data to fully support its main message and conclusions.

REVIEWERS' COMMENTS

Reviewer #2 (Remarks to the Author):

The authors have adequately addressed all of my previous concerns, I have no further comments.